# PD-1/PD-L1 checkpoint blockade harnesses monocyte-derived macrophages to combat cognitive impairment in a tauopathy mouse model

Neta Rosenzweig[1], Raz Dvir-Szternfeld[1,2], Afroditi Tsitsou-Kampeli[1], Hadas Keren-Shaul[2], Hila Ben-Yehuda[1], Pierre Weill-Raynal[1], Liora Cahalon[1], Alex Kertser[1], Kuti Baruch[1], Ido Amit [2], Assaf Weiner[2] & Michal Schwartz[1]

Alzheimer's disease (AD) is a heterogeneous disorder with multiple etiologies. Harnessing the immune system by blocking the programmed cell death receptor (PD)-1 pathway in an amyloid beta mouse model was shown to evoke a sequence of immune responses that lead to disease modification. Here, blocking PD-L1, a PD-1 ligand, was found to have similar efficacy to that of PD-1 blocking in disease modification, in both animal models of AD and of tauopathy. Targeting PD-L1 in a tau-driven disease model resulted in increased immuno-modulatory monocyte-derived macrophages within the brain parenchyma. Single cell RNA-seq revealed that the homing macrophages expressed unique scavenger molecules including macrophage scavenger receptor 1 (MSR1), which was shown here to be required for the effect of PD-L1 blockade in disease modification. Overall, our results demonstrate that immune checkpoint blockade targeting the PD-1/PD-L1 pathway leads to modification of common factors that go awry in AD and dementia, and thus can potentially provide an immunotherapy to help combat these diseases.

[1] Department of Neurobiology, Weizmann Institute of Science, Rehovot 7610001, Israel. [2] Department of Immunology, Weizmann Institute of Science, Rehovot 7610001, Israel. Correspondence and requests for materials should be addressed to M.S. (email: Michal.Schwartz@weizmann.ac.il)

Alzheimer's disease (AD) is a highly heterogeneous disease, in which several genetic risk factors have already been identified[1–4]. Yet, despite decades of research, therapies that individually target such identified risk factors have largely failed[5–9], suggesting that addressing single disease-associated factors, even critical ones, while possibly effective, is apparently insufficient for modifying the disease.

Over the last two decades, it became clear that systemic immune cells are important players in brain maintenance and repair, with implications to brain aging and neurodegenerative conditions[10–15]. Moreover, systemic immune deficiency has been associated with cognitive dysfunction[13], behavioral dysfunction[14] and reduced ability to cope with neurodegenerative conditions, including Amyotrophic lateral sclerosis (ALS)[16] and AD[17]. Accordingly, boosting recruitment of monocyte-derived macrophages to sites of brain pathology in several mouse models of AD, resulted in reduced brain pathology, in general, and reduced plaque burden, in particular[18–24].

We previously reported that recruitment of monocyte-derived macrophages is dependent on systemic availability of IFN-γ-producing CD4+ T cells[25,26]. In line with this finding, several independent studies have highlighted the negative role of overwhelming systemic immunosuppressive cells, or of immunosuppressive cytokines in AD pathology[26–28]. These results and others led us to envision that empowering the peripheral immune system would facilitate the recruitment of disease-modifying leukocytes to the brain parenchyma. Testing this premise in an amyloid-beta-driven AD mouse model, 5XFAD[29], led us to discover that transient reduction of systemic immune suppression (by reducing systemic levels of FoxP3 regulatory T cells or by blocking the inhibitory programmed-death (PD)-1 immune checkpoint pathway), could lead to Alzheimer's disease modification[26,30].

Here, we hypothesized that immune checkpoint blockade might activate common immune-dependent repair mechanisms, irrespective of the primary cause of the disease pathology. We found that targeting PD-1 or its PD-L1 ligand could modify AD pathology in a mouse model of amyloid pathology, 5XFAD, as well as in an animal model of tau pathology, expressing the human-tau gene with two mutations associated with frontotemporal dementia (DM-hTAU)[31]. In DM-hTAU mice, systemic administration of anti-PD-L1 blocking antibody mitigated both cognitive deficits as well as pathological manifestations of the disease, and modified the immunological milieu of the brain. Moreover, single-cell RNA-Seq revealed a unique reparative role of the infiltrating monocyte-derived macrophages, which substantiated their beneficial role in the anti-PD-L1 Alzheimer's disease therapy.

## Results

**Monthly treatment with anti-PD-1 antibody delays cognitive decline and supports neuronal rescue.** In our recent study, we showed that in male 5XFAD mice, administration of two injections of anti-PD-1 antibody (0.25 mg at a 3-day interval) resulted, 1 month later, in reversal of cognitive loss and modification of some of the pathological features of AD[30]. Here, we first repeated this experiment in female mice to ensure that the treatment is effective in both genders, and also tested if a single dose of 0.5 mg anti-PD-1 antibody could be as effective as two injections of 0.25 mg given in at 3-day interval[30]. To this end, 5XFAD female mice were treated with either two injections of 0.25 mg or with a single injection of 0.5 mg of anti-PD-1. Spatial learning/memory function was assessed 1 month later using the radial arm water maze (RAWM) task. Both treatment regimens provided a similar beneficial effect on cognitive performance, relative to treatment

with isotype-matched IgG2a control antibody (Supplementary Fig. 1). Therefore, for our subsequent experiments, we used a single injection of the antibody at the indicated dose, rather than a split dose, and worked with both male and female mice, as indicated throughout this study.

To examine whether the beneficial effect associated with PD-1 blockade could delay cognitive decline, we treated a cohort of 5XFAD mice (female and male, in equal proportions in all tested groups; approximately 3.5 months of age) with anti-PD-1 antibody, and continued the treatment once a month for an additional 2 months (a single injection of 0.5 mg once every 4 weeks) (Fig. 1a). We evaluated these mice for their spatial learning and memory performance twice, first at 5.5 months and subsequently at 6.5 months of age (changing the cues for the second RAWM test). Our results revealed that while at the age of 5.5 months, the control IgG-treated animals showed residual learning/memory skills in this maze, and thus exhibited some learning on the last trial of the second day (Fig. 1b), by the age of 6.5 months, disease progression was further manifested in these IgG-control-treated mice, with an additional decrease in functional performance (Fig. 1c, d). In contrast, the anti-PD-1-treated group retained learning capacity. Figure 1d illustrates the decline in the performance of the IgG-treated mice between the average ages of 5.5 and 6.5 months, in contrast to the anti-PD-1 antibody-treated group. Notably, some animals in all groups showed motor deficits and could not complete the swimming task; these animals were not included in the behavioral analysis, regardless of the treatment group, but were included in the pathological and histological studies, as detailed below.

In AD, neuronal loss was reported to correlate with impairment in spatial learning/memory skills[32–34]. We evaluated neuronal survival in the brains of these mice, after the final behavioral test, using Cresyl violet staining[35,36]. This staining revealed a reduction in large pyramidal neurons in both the cortex and subiculum, relative to age-matched wild-type (WT) littermates (Fig. 1e–h), as previously reported[35]. However, a significantly higher number of surviving cells was found in the anti-PD-1-treated 5XFAD mice relative to the IgG control-treated group (Fig. 1e–h). Neuronal survival was also assessed by immunostaining for Neu-N, a nuclear marker of mature neurons, revealing a higher number of Neu-N immunoreactive cells in the subiculum of the anti-PD-1-treated mice compared to the IgG control group (Fig. 1i, j). These results suggested that treatment with PD-1 contributed to neuronal protection/rescue. Importantly, when we included in the analysis mice that failed to complete the cognitive task due to the motor deficits, the results were not altered, and remained just as robust (Supplementary Fig. 2).

The molecular mechanisms underlying neuronal loss in AD have been previously linked to neuronal Caspase-3 activation, which is involved in the apoptotic death of these cells[35,37]. We therefore examined the subiculum of the same brains to determine whether the treatment with anti-PD-1 was associated with a reduction in the levels of activated Caspase-3 in Neu-N+ cells. We found that mice treated with anti-PD-1 antibody showed a reduction in activated Caspase-3 immunoreactivity in Neu-N+ neurons, as compared to the IgG-treated group (Fig. 1k, l).

**Blockade of PD-L1 in 5XFAD mice mitigates cognitive deficits and cerebral pathology.** PD-1 can interact with both PD-L1 and PD-L2[38]. To test a more specific inhibitor of these checkpoint pathways, we used PD-L1 blockade. Notably, PD-L1 is expressed by several cell types, including myeloid cells, epithelial cells and regulatory T cells[38]. We compared the blocking effect of PD-L1 and PD-1 on disease manifestations (a scheme depicting the

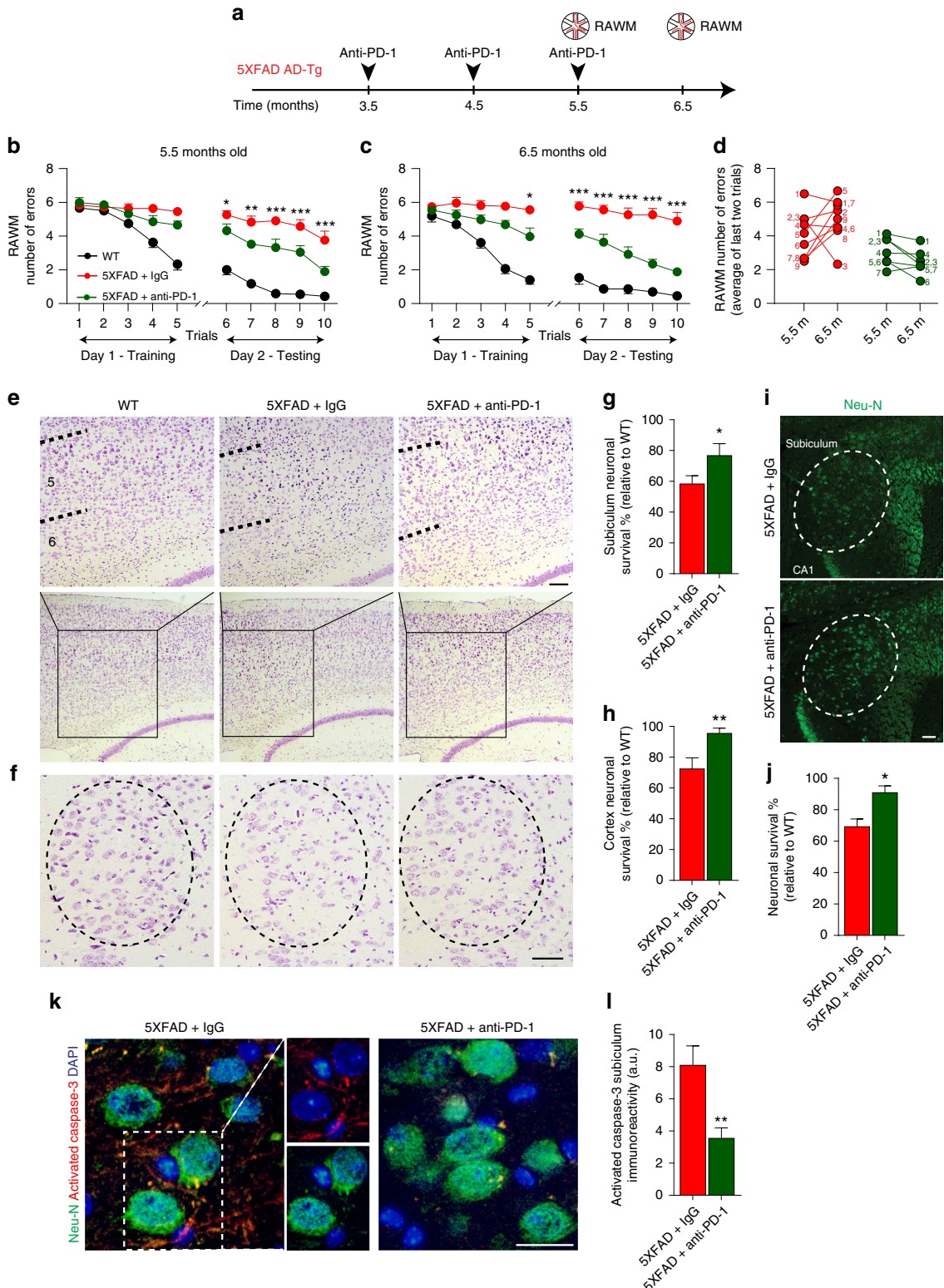

experimental design for all antibodies tested in this set of experiments is shown in Fig. 2a). In the first experiment, 6-month-old 5XFAD mice were treated with a single dose of anti-PD-L1 antibody (0.1, 0.5, or 1.5 mg per mouse), injected intraperitoneally. Mice were assessed 1 month later, using RAWM. While 0.1 mg of anti-PD-L1 did not have any effect on cognitive performance compared to treatment with IgG2b, the relevant isotype control, both 0.5 and 1.5 mg doses had a similar beneficial effect; the higher dose was slightly superior, though the difference

was not significant (Fig. 2b). To compare the efficacy of blocking PD-1 or blocking PD-L1, we performed an additional experiment in which we subjected 5XFAD mice to either a single injection of 0.5 mg of anti-PD-1 antibody or anti-PD-L1 antibody. We found that anti-PD-L1 antibody treatment was similarly effective as the anti-PD-1 antibody in altering cognitive outcome (Fig. 2c). The two groups of mice treated with IgG2a or IgG2b isotype-matched control antibodies for the anti-PD-1 and anti-PD-L1 groups, respectively, gave similar results in the behavioral studies, and

**Fig. 1** Early treatment with anti-PD-1 antibody delays cognitive loss and rescues neurons in 5XFAD mice. 5XFAD mice were treated with either PD-1–specific antibody or IgG control starting at an average age of 3.5 months, and continued once a month (total of three injections). Experimental design is presented in (**a**). Black arrows indicate time points of treatment. Time points of cognitive testing are also indicated. **b** RAWM performance of 5.5-month-old anti-PD-1-treated mice ($n = 7$), control antibody (IgG)-treated 5XFAD mice ($n = 9$), and wild-type (WT) animals ($n = 8$). **c** RAWM performance of the same mice one month later (6.5-month-old). (Two-way repeated-measures ANOVA and Dunnett's post-hoc test for multiple comparisons between the two treated groups.) **d** Comparison of the performance of anti-PD-1 and IgG-treated groups at 5.5 and 6.5 months; values indicating the number of errors for each mouse, taken from the last measurement on the second day of the test. **e, f** Representative Cresyl violet-stained images of the cortex and subiculum, and **g, h** quantitative analysis of IgG-treated 5XFAD mice ($n = 9$) as compared to anti-PD-1-treated 5XFAD mice ($n = 7$); the WT ($n = 7$) controls that were analyzed were used to calculate the percentage of surviving neurons. **i, j** Images and quantitative analysis of Neu-N$^+$ neurons in the subiculum of anti-PD-1-treated 5XFAD mice ($n = 7$), and IgG-treated 5XFAD mice ($n = 9$), respectively; the WT controls ($n = 7$) that were analyzed were used to calculate the percentage of surviving neurons. Neuronal survival is presented relative to the number of the pyramidal neurons in the age-matched WT littermates. **k, l** Representative immunofluorescence images and quantitative analysis of Neu-N$^+$ neurons (green) in the subiculum, showing increased immunoreactivity of Activated Caspase-3 (red) in IgG-treated 5XFAD mice ($n = 9$), in comparison to anti-PD-1-treated 5XFAD mice ($n = 7$) (Student's *t* test). Scale bars, 50 µm (**e, f**), 100 µm (**i**), and 25 µm (**k**). Data are presented as mean ± s.e.m.; *$P < 0.05$, **$P < 0.01$, ***$P < 0.001$

were combined in the final analysis. At the end of the behavioral studies, brains were removed after perfusion and were used to assess effects on pathological and morphological aspects of the disease by immunohistochemistry. The results revealed a similar reduction in plaque burden (measured by the number of plaques and by the area covered by the plaques) in the dentate gyrus (DG) and cortex, and in gliosis (measured by Glial Fibrillary Acidic Protein (GFAP)-immunoreactivity) in both anti-PD-1 and anti-PD-L1 treatment groups, as compared to the IgG-treated group (Fig. 2d–g). In the results presented in Fig. 2c–g, animals that showed motor deficits, irrespective of the treatment, were excluded from the behavioral analyses, and from the histology presented in Fig. 2d–g. Notably, however, the effect of PD-1 and PD-L1 blockade, on plaque pathology and GFAP, was also significant when those animals that exhibited motor deficits were included in the analysis of the effect on brain pathology (Supplementary Fig. 3). Brain sections from this experiment were also used to evaluate the effect on the presynaptic marker, synaptophysin[29]. We observed a higher immunoreactivity specific to synaptophysin in both the anti-PD-1- and anti-PD-L1-treated groups relative to mice treated with IgG (Fig. 2h, i). We also determined the effects of PD-1 or PD-L1 blockade on the cytokine milieu of the hippocampi of the treated mice using real-time (RT)-qPCR. The results showed reduced *il-12p40* levels, and elevation of *il-10* relative to the IgG-treated 5XFAD mice (Fig. 2j, k), in the hippocampi of mice treated with either anti-PD-1 or anti-PD-L1.

We continued with anti-PD-L1, and directly examined whether the treatment could reverse functional loss. To this end, we used a new cohort of 5XFAD mice, testing their cognitive performance prior to the antibody administration, and 1 month following treatment (Fig. 2l–n). Before the treatment was administered, all tested mice showed cognitive impairment in the RAWM at 5.5 months of age (Fig. 2m). These mice were then divided into two treatment groups; one group received anti-PD-L1 and the other group received the IgG control antibody (a single injection of 1.5 mg per mouse). One month after the treatment, a robust and significant behavioral improvement relative to their baseline performance was found in the group treated with anti-PD-L1, but not in the IgG-treated group (Fig. 2n), suggesting that blocking an inhibitory checkpoint could reverse functional loss, at least at the disease stages tested.

**Blockade of the PD-1/PD-L1 axis in a mouse model of tau pathology mitigates cognitive deficits and reduces inflammatory cytokines in the brain.** Since treatment using immune checkpoint blockade targets systemic immune cells, we hypothesized that it would also be effective in mouse models driven by tau pathology. To test whether a treatment that targets PD-1/PD-L1 blockade could be effective in tauopathy, we used a

transgenic mouse model that expresses two mutations of the human-tau gene (K257T/P301S; double mutant, DM-hTAU) associated with Frontal-Temporal dementia[31]. Pathological features in these mice include cognitive deficits, neuroinflammation, glial cell activation, and phosphorylation of tau proteins within the brain, and accumulation of aggregated tau protein[31]. We first treated DM-hTAU mice at 8 months of age (an age in which these mice show a pronounced cognitive deficit) with anti-PD-1 or anti-PD-L1, using the same dose of 0.5 mg, as was used to compare these two antibodies in the 5XFAD mice (Fig. 3a); isotype-matched antibodies (IgG2a, IgG2b) were used as negative controls. Both female and male animals were included and equally distributed in all groups. We assessed cognitive performance using T-maze to evaluate short-term spatial memory (Fig. 3b–d), an assay that is commonly used in studies of of the DM-hTAU animal model[31,39,40]. Figure 3c shows the time spent in the three arms for each animal in all groups, which helps illustrate the preference for the novel arm. The two groups that received either anti-PD-1 or anti-PD-L1 antibodies demonstrated an increased preference for the novel arm relative to the two other arms in the T-maze assay as compared to the IgG-treated control group, 1 month after a single injection of the 0.5 mg antibodies (Fig. 3b, c). Note that the results obtained with the two IgG antibodies, the relevant isotype controls for the tested anti-PD-1 and anti-PD-L1 antibodies, were combined in Fig. 3c, as they gave the same results, shown by their separate analysis in Supplementary Fig. 4. Notably, hereafter, for simplicity, in all the figures that show results of T-maze measurement we presented only the preference for the novel arm.

Separately analyzing the female and male animals that were tested in the T-maze task in this experiment revealed a similar trend for each gender (Supplementary Fig. 5). We continued with the male in this experiment and tested their performance in Y-maze task, which measures spontaneous alternations. We found that both anti-PD-1 and anti-PD-L1 treatment groups exhibited improved performance compared to the IgG-treated control group (Fig. 3d). Age-matched WT littermates were assessed as an additional control for normal learning/memory performance in the T- and Y-maze tasks. Following the last behavioral test of all mice, we excised the brains and tested whether the beneficial effect of PD-1/PD-L1 blockade was associated with changes in the brain's inflammatory cytokine profile. Quantitative RT-qPCR revealed that both anti-PD-1 and anti-PD-L1 treatment led to a reduction in the expression levels of the inflammatory cytokines *tnf-a, il-6, il-12p40, and Il-1β* relative to the IgG-treated control group (Fig. 3e–h).

To better understand the mechanism that leads to immunomodulation within the brain parenchyma induced by targeting PD-L1, we used a new cohort of DM-hTAU mice, and tested

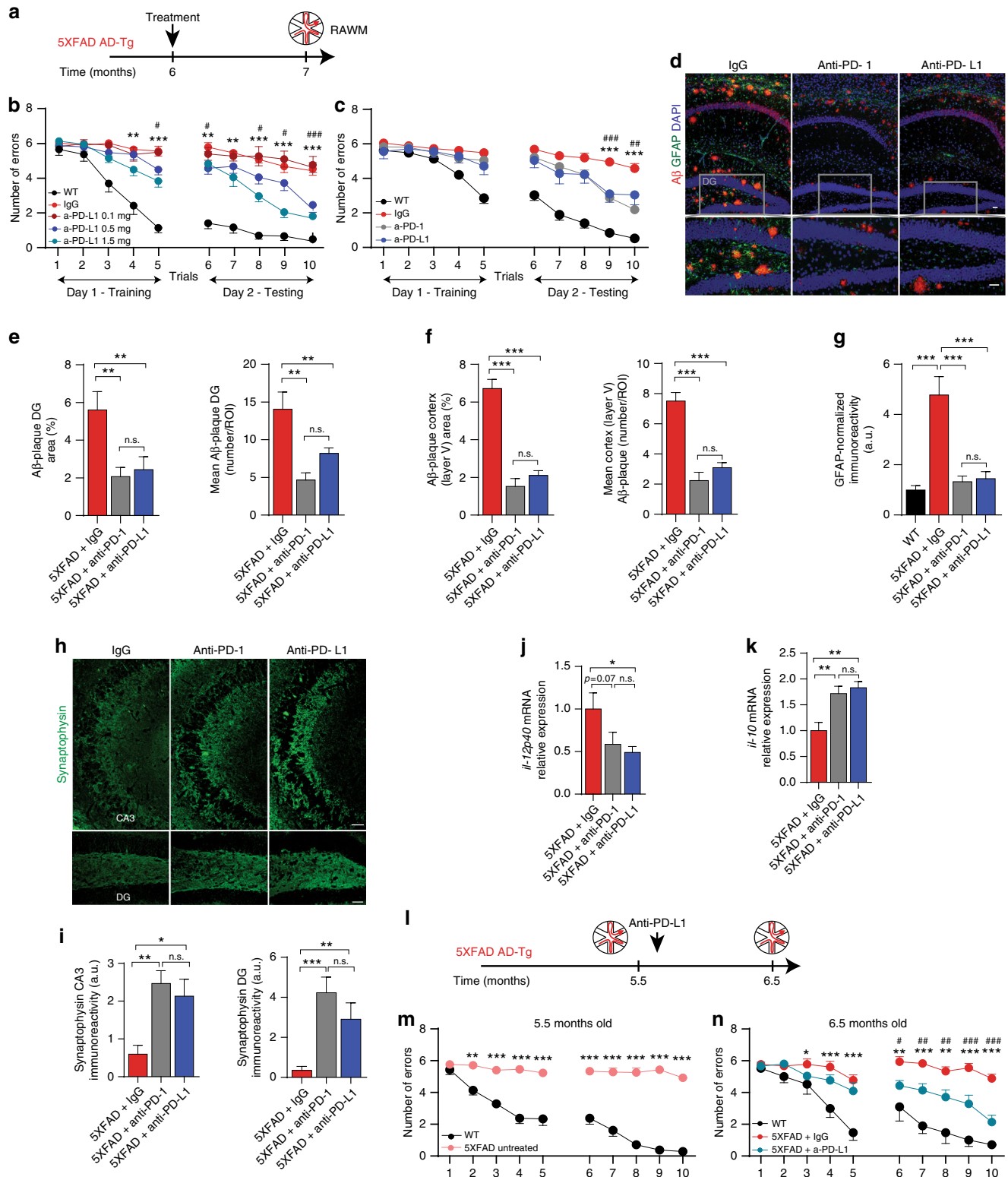

whether the efficacy of anti-PD-L1 is dependent on the antibody dosage in these animals. To this end, DM-hTAU mice were treated with a single injection of anti-PD-L1 at 0.1, 0.5, or 1.5 mg/mouse, relative to a single injection of 1.5 mg/mouse IgG2b control antibody. WT littermates were used as a control for intact cognitive performance. Results are presented as the time spent in the novel arm, revealing that 0.1 mg of anti-PD-L1 antibody was ineffective, while dosages of 0.5 or 1.5 mg were both effective in improving cognitive performance (Fig. 3b–d). Treatment with

anti-PD-L1 antibody at a dose of 1.5 mg/mouse was slightly more effective than the 0.5 mg dose, but the difference between these two dosages was not significant (Fig. 3i). Wild-type age-matched mice from the DM-hTAU colony were also tested for the effect of PD-L1 antibody on their cognitive response to the treatment by T-maze; no effect was observed (Supplementary Fig. 6).

Elevation in brain-parenchymal levels, of the proinflammatory cytokine IL-1β was previously linked to cognitive decline in murine models of dementia[41–43]. Therefore, we further focused

**Fig. 2** Blockade of PD-L1 overcomes loss of cognitive performance and reduces cerebral pathology in 5XFAD mice. 5XFAD mice were treated with either anti-PD-1, anti-PD-L1 or IgG. Experimental design is presented in **a**. **b** RAWM performance of WT littermates ($n = 10$), and of mice treated with 0.1 mg/mouse ($n = 8$), 0.5 mg/mouse ($n = 9$), or 1.5 mg/mouse ($n = 9$), of anti-PD-L1 antibody, or with 1.5 mg/mouse IgG ($n = 10$). **c** RAWM performance of WT ($n = 19$), and of 5XFAD mice treated with either 0.5 mg of anti-PD-1 antibody ($n = 14$), or with 0.5 mg of anti-PD-L1 antibody ($n = 7$); IgG control antibody ($n = 6$ IgG2a; $n = 9$ IgG2b). Two-way repeated-measures ANOVA and Dunnett's post-hoc test for multiple comparisons between each anti-PD-1-treated group and the IgG-treated groups (*$P < 0.05$, **$P < 0.01$, ***$P < 0.001$) and between Anti-PD-L1 and IgG-treated groups (#$P < 0.05$, ##$P < 0.01$, ###$P < 0.001$) . Results shown are pooled from two experiments. **d–g** Analysis of disease pathology from a single experiment; one hemisphere of each brain was taken for histology and one for quantitative measurements of mRNA. **d** Representative immunofluorescence images of brains stained for Aβ (in red), GFAP (in green) and DAPI nuclear staining (scale bars, 100 μm). **e**, **f** Quantitative analysis of Aβ in anti-PD-1-treated mice ($n = 9$), anti-PD-L1-treated mice ($n = 7$), and IgG-treated ($n = 9$) mice. **g** GFAP in anti-PD-1-treated ($n = 9$), anti-PD-L1-treated mice ($n = 7$), IgG-treated ($n = 8$) mice, and WT ($n = 6$). Plaque area and numbers were quantified in the dentate gyrus (DG) and in the cerebral cortex (layer V), and GFAP immunoreactivity was measured in the hippocampus (one-way ANOVA and Fisher's exact test). **h**, **i** Representative immunofluorescence images, and quantitative analysis of synaptophysin in the hippocampal CA3 and in the DG, in the brains of anti-PD-1-treated mice ($n = 6$), anti-PD-L1-treated 5XFAD mice ($n = 7$), and IgG-treated mice ($n = 9$) (one-way ANOVA and Fisher's exact test). Scale bars, 50 μm (**h**). **j**, **k** RT-qPCR, in hippocampal tissue isolated from mice treated with IgG control ($n = 5$), anti-PD-1 antibody ($n = 5$), or anti-PD-L1 antibody ($n = 5$) (one-way ANOVA and Fisher's exact test). **l** Experimental design of the mice studied in **m** and **n**. **m** RAWM performance before ($n = 13$) treatment, and of the same mice (**n**) following their treatment either with 1.5 mg/mouse ($n = 7$) anti-PD-L1, or with IgG control ($n = 6$). WT littermates were tested twice ($n = 7$) (two-way repeated-measures ANOVA and Dunnett's post-hoc test for multiple comparisons between anti-PD-L1 IgG-treated groups). **b**, **c**, **e–g**, **i–k** mean ± s.e.m.; *$P < 0.05$, **$P < 0.01$, ***$P < 0.001$. DAPI: 4′,6-diamidino-2-phenylindole, RT-qPCR: real-time quantitative PCR

on IL-1β in DM-hTAU mice. Analysis of brain sections from DM-hTAU mice, treated with either anti-PD-L1 antibody or IgG2b control antibody, revealed IL-1β immunoreactivity in the brain parenchyma, the levels of which decreased following treatment with anti-PD-L1 antibody as compared with the IgG2b control (Fig. 3j). Moreover, IL-1 β immunoreactivity was found to be colocalized mainly with GFAP-expressing cells, a marker of astrocytes, but not with the myeloid cell marker IBA-1, including microglia and monocyte-derived macrophages (Fig. 3k).

To quantify the effect of anti-PD-L1 on IL-1β levels in the brains of the treated DM-hTAU mice using an immunoassay, we repeated the experiment using four groups of mice, including DM-hTAU and WT littermates that were left untreated, and compared them with DM-hTAU treated with IgG2b control antibody or treated with anti-PD-L1 antibody. After verifying the effect of the anti-PD-L1 blockade on behavior (Supplementary Fig. 7), which was tested 1 month following treatment initiation, we perfused the mice and excised their brains; half of each brain was used to prepare separate protein extracts from hippocampi and cortices (see Methods), while the other hemisphere from each brain was processed for histochemistry (Fig. 4f, g). We found significantly higher protein levels of IL-1β in the hippocampi of DM-hTAU mice relative to age-matched wild-type littermates, and a significant reduction following anti-PD-L1 antibody treatment relative to untreated, or IgG2b-treated DM-hTAU mice (Fig. 3l). Moreover, linear regression analysis revealed a correlation (Fig. 3m) between behavioral outcome shown for these mice in Supplementary Fig. 7, and levels of IL-1β in the same mice shown in Fig. 3l. Finally, to assess the robustness of PD-L1 blockade on disease modification, we tested an additional anti-PD-L1 antibody, a human anti-PD-L1 clone that cross-reacts with the mouse PD-L1 ligand[44] (see Methods); this antibody is directed against a human PD-L1 epitope and has a slightly lower affinity to the mouse PD-L1 than the rat-anti-mouse PD-L1 antibody that was used throughout this study. Treatment of DM-hTAU mice with this antibody elicited a similar effect on cognitive performance improvement (Fig. 3n).

**Blockade of the PD-1/PD-L1 axis in a mouse model of tau pathology reduces pathology in the brain**. Neuroinflammation in murine models of tauopathies was previously shown to be associated with enhanced tau hyperphosphorylation, which was found to accelerate disease progression[41,45,46], and to

increase levels of aggregated tau, as well. We tested brain sections from the experiment described in Fig. 3b–d to determine whether the reduced inflammatory profile attained as a result of the treatment with anti-PD-1 and anti-PD-L1 would be manifested by reduced tau hyperphosphorylation, assessed using immunohistochemistry. Immunohistochemical analysis of brain sections from DM-hTAU mice revealed reduced immunoreactivity in staining for the AT-100 (Phospho-Tau Thr212, Ser214) and AT-180 (Phospho-Tau Thr231) epitopes, in the hippocampal CA1 and CA3 regions, following anti-PD-1 and anti-PD-L1 blockade, as compared to the IgG isotype control group (Fig. 4a–d).

For quantifying the effect of anti-PD-L1 on tau aggregation, we used the protein extracts from hippocampi as well as from the cortices of the DM-hTAU mice, described above; the cognitive performance of these animals is shown in Supplementary Fig. 7. Using an immunoassay for tau aggregation, we found a significant reduction in cortical and hippocampal tau aggregation in DM-hTAU mice treated with anti-PD-L1 antibody as compared with IgG control-treated group, relative to untreated DM-hTAU mice (Fig. 4e).

Finally, neuronal survival was analyzed by Cresyl violet staining in cortical layer 5, using sections of DM-hTAU mouse brains; sections were taken from the mice that were randomly selected from the experiments described in Fig. 3d and in Supplementary Fig. 7. A higher number of neurons survived in the anti-PD-L1-treated animals relative to the IgG2b-treated group (Fig. 4f, g).

**Targeting the PD-1/PD-L1 pathway in a mouse model of tau pathology enhances recruitment of monocyte-derived macrophages to the brain parenchyma**. In both 5XFAD and J20 mouse models of AD, disease progression is associated with a reduction of CP expression of leukocyte-trafficking molecules[26,47]. Treatment with anti-PD-1 antibodies results in 5XFAD mice in enhanced recruitment of monocyte-derived macrophages to the brain[31]. These findings, together with our current results demonstrating a beneficial effect of anti-PD-L1 in the DM-hTAU model of dementia (Figs. 3 and 4), prompted us to test whether the observed beneficial effect of targeting PD-L1 on cognitive function and disease pathology in this tau mouse model was also associated with enhanced trafficking of immune cells to the diseased brain. To this end, we first tested whether the administration of antibody directed against PD-L1 induced elevation of effector memory T cells in DM-hTAU mice. Analyzing the

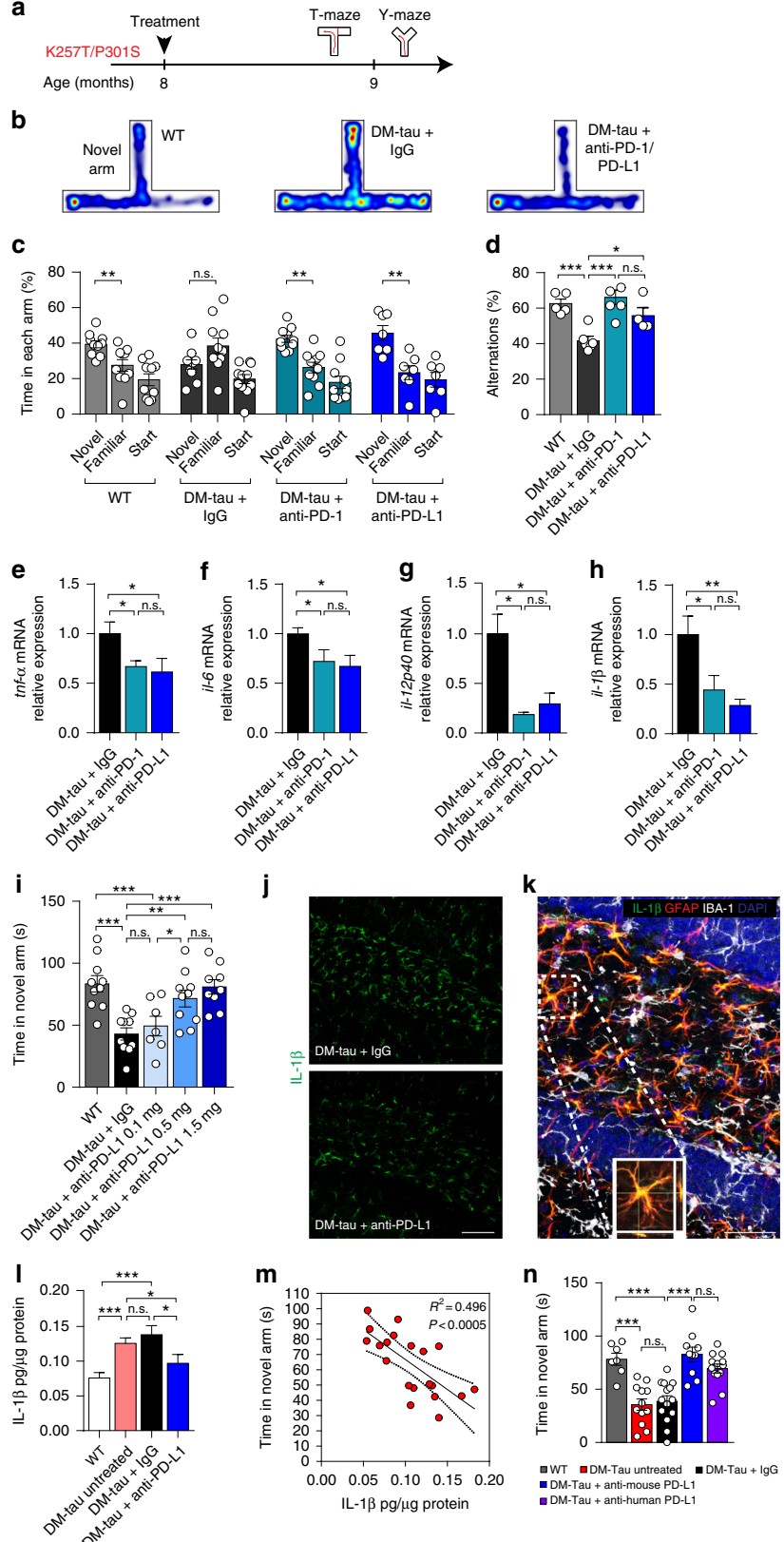

spleens of DM-hTAU mice 2 weeks after the anti-PD-L1 antibody administration revealed increased levels of effector memory T cells (T$_{EM}$; CD44$^+$CD62L$^{low}$) relative to those in IgG-treated mice (Fig. 5a, b), as evaluated by flow cytometry analysis. We further analyzed, by flow cytometry, in the DM-hTAU mice whether the treatment facilitated recruitment of monocyte-

derived macrophages (CD45$^{high}$CD11b$^{high}$) to the brain parenchyma. We found a significant increase in CD45$^{high}$CD11b$^{high}$ cells in the brains of DM-hTAU mice treated with anti-PD-L1 antibody relative to those treated with the IgG2b isotype control (Fig. 5c, d). To confirm the lineage of these cells, which we classified as mainly monocyte-derived macrophages based on

**Fig. 3** Blockade of the PD-1/PD-L1 axis in DM-hTAU mice reduces cognitive deficits and cerebral pathology. DM-hTAU mice (8-month-old) were treated with anti-PD antibody, anti-PD-L1 antibody, or IgG control antibody (0.5 mg/mouse); experimental design is presented in **a**. **b** Representative heat-map plots of the time spent in the distinct arms, by the three tested groups. **c** Effect of anti-PD-1/PD-L1 antibodies on performance in T-maze. DM-hTAU mice treated with anti-PD-1 antibody ($n = 10$) or anti-PD-L1 antibody ($n = 7$) exhibited increased novel arm preference, compared to IgG controls ($n = 11$); age-matched WT littermates ($n = 10$) were also tested. **d** Y-maze performance of male DM-hTAU mice treated with anti-PD-1 antibody ($n = 6$), anti-PD-L1 antibody ($n = 4$), or IgG isotype control ($n = 6$), and age-matched WT ($n = 5$). **e**-**h** Hippocampi were isolated from one hemisphere of the brains of the animals in **d** for quantitative RT-qPCR; IgG control ($n = 6$), anti-PD-1 antibody ($n = 6$), or anti-PD-L1antibody ($n = 4$) (one-way ANOVA and Fisher exact test). **i** Dose-dependent effect of anti-PD-L1 antibody on T-maze performance of DM-hTAU mice (male and female; 9-month-old) (0.1 mg/mouse ($n = 7$), 0.5 mg/mouse ($n = 10$), or 1.5 mg/mouse ($n = 9$), and IgG control at 1.5 mg/mouse ($n = 10$)), WT ($n = 10$). Results are expressed as the time spent in the novel arm (one-way ANOVA and Fisher's exact test). **j** Representative images of IL-1β immunoreactivity in mice treated with anti-PD-L1 antibody or with IgG control. **k** Representative orthogonal projection of confocal z-axis stacks, showing colocalization of IL-1β (green) with GFAP+ astrocytes (red), but not with IBA-1+ microglia/macrophages (white), in the dentate gyrus. Cell nuclei (DAPI, blue). Scale bar, 100 μm. **l** Immunoassay of hippocampal IL-1β protein levels in untreated mice ($n = 3$), mice treated with anti-PD-L1 antibody ($n = 6$) or IgG ($n = 6$) and WT littermates ($n = 6$), normalized to mg protein in each homogenate. **m** Correlation between cognitive performance in the T-maze, shown in Supplementary Fig. 7, and IL-1β protein levels for all groups. **n** T-maze of DM-hTAU mice (9-month-old, male and female) treated with either 1.5 mg of anti-mouse PD-L1 antibody ($n = 11$), or with anti-human PD-L1 antibody ($n = 13$), 1.5 mg/mouse of IgG control ($n = 14$), and compared to untreated DM-hTAU mice ($n = 12$) and WT littermates ($n = 7$). One-way ANOVA and Fisher exact test. In all panels, error bars represent mean ± s.e.m.; *$P < 0.05$, **$P < 0.01$, ***$P < 0.001$

their high expression of CD45 and CD11b, we repeated this experiment with bone marrow (BM)-chimeric mice, in which the donor BM cells were taken from mice with GFP-labeled hematopoietic cells[48]. To create such chimera, recipient DM-hTAU mice were conditioned with lethal-dose irradiation, with the radiation beam targeting the lower part of the body while avoiding the head, prior to BM transplantation[49]. Following establishment of chimerism (see Methods), animals were treated with either anti-PD-L1 antibody or with control IgG2b. Analysis of the brains 2 weeks after the administration of the antibody, by flow cytometry, revealed that among the CD45highCD11bhigh cells, about 50% of the cells were GFP+, which was consistent with the extent of the chimerism, and confirmed their identity as infiltrating monocytes, rather than activated resident microglia (Fig. 5e, f). No GFP+ cells were seen among the CD45lowCD11b+ cells. Notably, we gated only on GFP+CD45+CD11b+ myeloid cells; BM-derived cells that were GFP+CD45+CD11b- were not analyzed. Treatment with anti-PD-L1 antibody resulted in an approximately threefold increase in the frequency of GFP+CD45highCD11bhigh cells, relative to IgG2b-treated control (Fig. 5f). Notably, this number underestimates the number of homing macrophages, since the chimerism was about 50%. The brains from other mice from the same experiment were excised and processed for immunohistochemistry, which revealed the presence of GFP+IBA-1+ myeloid cells in the cortex of the anti-PD-L1-treated mice (Fig. 5g). We also stained brain sections from the same animals for the anti-inflammatory cytokine, IL-10, and observed its colocalization with infiltrating monocyte-derived macrophages, but not with IBA-1+GFP- microglia (Fig. 5h).

The overall number of monocyte-derived macrophages that infiltrated the brain was low, and the number of those that were GFP+ was even lower. Therefore, we further characterized the infiltrating cells by single-cell RNA-seq. We sorted from both IgG2b-treated and anti-PD-L1-treated groups all the CD45highCD11bhigh cells, thereby enriching the monocyte-derived macrophages within the analyzed samples. Clustering analysis of 899 cells (Supplementary Fig. 8) revealed that the infiltrating monocyte-derived macrophages were heterogeneous, and most likely included several activation states (as seen in clusters 5–10); clusters 1–4 represent activated microglia in several states, and clusters 11–12 indicate neutrophils. Analysis of differential genes in each cluster highlighted a unique signature displayed mainly by clusters 5 and 6, distinct from the resident homeostatic or activated microglia (clusters 1–4); the unique signature was manifested by expression of several molecules that could potentially mediate an important function in disease modification (Fig. 5i, j). One such uniquely expressed molecule is

the macrophage scavenger receptor 1 (*Msr1*) (also known as SRA1, SCARA1, or CD204), an important phagocytic receptor required for engulfment of misfolded and aggregated proteins[50,51], and found previously by us to be expressed by M2-like infiltrating monocyte-derived macrophages that are needed for spinal cord repair[52]. Notably, these macrophages expressed additional relevant functional molecules, among which are the insulin-like growth factor-1 (*igf1*) that was previously reported to enhance neurogenesis in the aged brain[53], lymphatic endothelium-specific hyaluronan receptor (*lyve1*) and the scavenger receptor stabilin-1 (*Stab-1*) (Fig. 5j), both of which are markers of anti-inflammatory macrophages, associated with wound healing and lymphogenesis[54]. Additional genes, found here to be uniquely expressed by infiltrating monocyte-derived macrophages, are scavenger receptors such as the sialic acid binding Ig-like lectin 1 (*Siglec1*) and the mannose receptor C-type (*Mrc1*) (Fig. 5j).

In light of the reported role of MSR1 in neurodegenerative diseases, we further focused on this scavenger receptor. Using immunohistochemistry, we confirmed the expression of MSR1 by the GFP+ (infiltrating) cells (Fig. 5k, l), in line with our previous findings[30]. Finally, to gain insight into the functional impact of MSR1-expressing macrophages on the repair process, we created BM chimeric DM-hTAU mice, in which the BM of the recipients mice was replaced with donor BM taken from MSR1-deficient mice. As controls we used DM-hTAU chimeric mice in which the recipients BM was replaced with BM taken from non-transgene wild-type littermates. Two weeks following the chimerism, the mice were examined for cognitive performance using the T-maze task. We also tested WT chimeric mice that received either wild-type BM or BM from MSR1$^{-/-}$ mice (Fig. 5m, n). Following the behavioral test, each group of DM-hTAU chimeric mice was divided into two groups that received either anti-PD-L1 antibody or the control IgG2b, and 4 weeks later were tested again for their performance in the T-maze. Another group of nonchimeric DM-hTAU littermates that received IgG2b control was also evaluated. Anti-PD-L1 antibody reversed cognitive loss in DM-hTAU chimeras harboring BM from wild-type mice, while DM-hTAU chimeras harboring MSR1$^{-/-}$ BM lost the ability to respond to PD-L1 blocking antibody and failed to show improved cognitive ability (Fig. 5n).

Taken together, our results suggest that systemic immune activation, under conditions of chronic neuroinflammation, associated with murine models of tauopathies, facilitates the homing of monocyte-derived macrophages to the diseased brain and that these cells are key players in the anti-PD-L1 effect on disease modification.

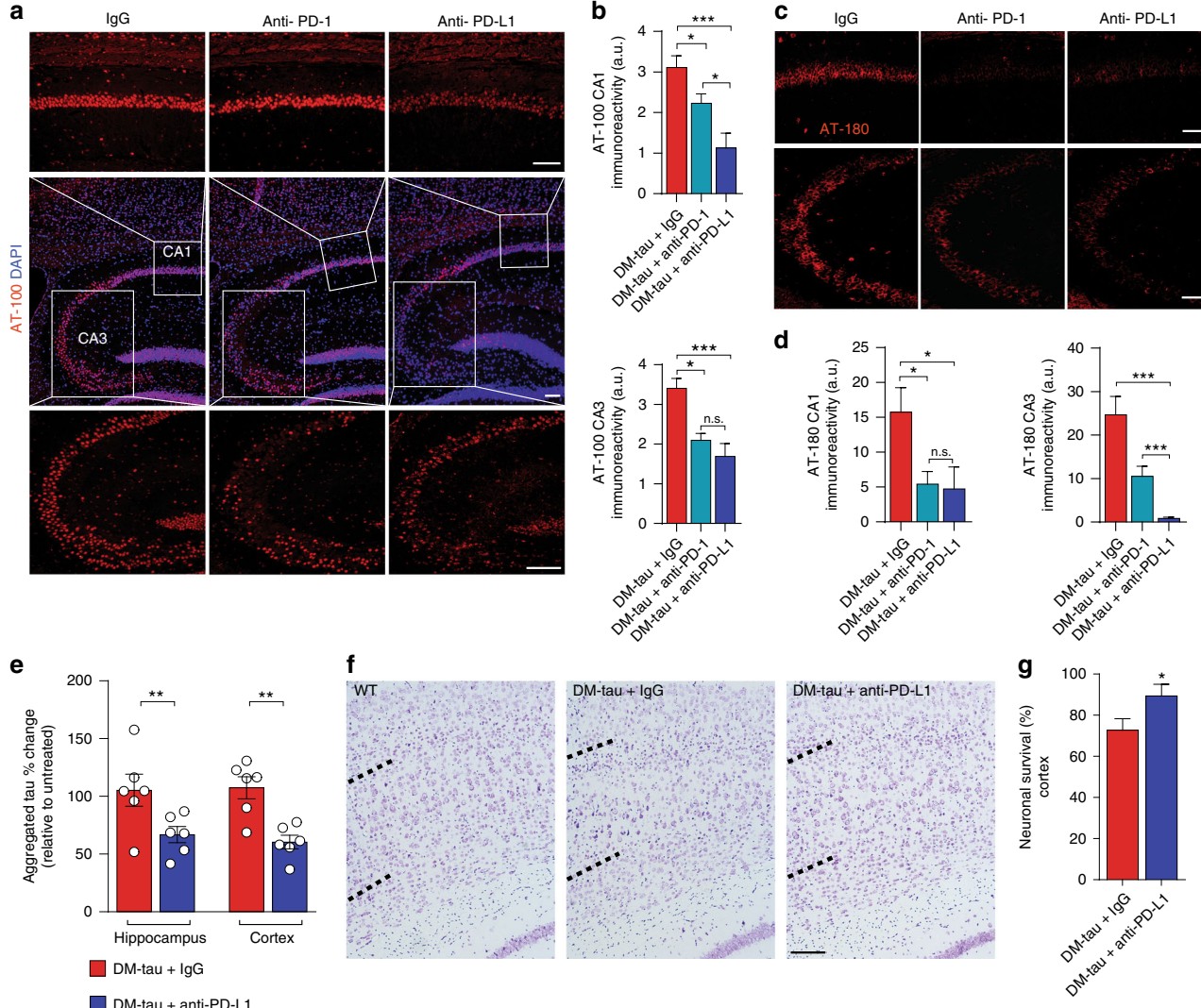

**Fig. 4** Blocking PD-1/PD-L1 pathway reduces tau hyperphosphorylation in DM-tau mice. Immunostaining of tau-phosphorylation in brains of 8-month-old male DM-hTAU mice excised 1 month after treatment with anti-PD-1, anti-PD-L1, or isotype-matched control antibody (IgG). The analyzed mice were taken from the experiment in Fig. 3d (**a–d**). **a** Representative immunofluorescence images, and **b** quantitative analysis of AT-100. Immunoreactivity of AT-100 was measured in the hippocampal CA1 and CA3 regions of DM-hTAU + IgG ($n = 11$), DM- hTAU + anti-PD-1 ($n = 10$), and DM-tau + anti-PD-L1 ($n = 4$). **c, d** Representative immunofluorescence images, and quantitative analysis of AT-180 staining. Immunoreactivity of AT-180 was measured in the hippocampal CA1 and CA3 regions of the following treatment groups DM-hTAU + IgG ($n = 11$), DM-hTAU + anti-PD-1 ($n = 10$), DM-hTAU + anti-PD-L1 ($n = 4$) (one-way ANOVA and Fisher's exact test). Scale bars, 100 μm (**a, c**). **e** Hippocampal and cortical levels of aggregated tau, measured in the protein extracts of the mice shown in Supplementary Fig. 7. Results are normalized to mg protein, and are expressed relative to untreated DM-hTAU; $n = 6$ per group (Student's $t$ test). **f, g** Representative Cresyl violet-stained images of the cortex and quantification of cortical layer five neurons, of male and female DM-hTAU mice (average cohorts aged 9 months) treated with anti-PD-L1 ($n = 6$) compared to IgG control group ($n = 7$); the wild-type ($n = 6$) controls that were analyzed were used to determine the percentage of surviving neurons. (Student's $t$ test). Scale bars, 100 μm. In all panels, error bars represent mean ± s.e.m.; *$P < 0.05$, **$P < 0.01$, ***$P < 0.001$

## Discussion

In the present study, we show that revitalizing systemic immunity using antibodies that block either PD-1 or its ligand, PD-L1, could modify brain pathology and restore cognitive performance in a mouse model of tauopathy, in addition to its effect in a beta-amyloid AD model, and that this effect is mediated, at least in part, via recruitment of monocyte-derived macrophages.

We found that in the 5XFAD mouse model, the beneficial effect of systemic administration of antibodies directed against PD-1 on cognitive function is associated with rescue of neurons, reduced neuronal apoptosis, and reduced synaptic loss. We further found that administration of antibodies directed against the PD-1 ligand, PD-L1, had a similar effect as that of anti-PD-1 in both 5XFAD mice and in the DM-hTAU animal model.

Analyzing the underlying immunological mechanisms in the DM-hTAU mouse, using anti-PD-L1 antibody, revealed elevated levels of effector memory T cells in the periphery, which was followed by an increased number of monocyte-derived macrophages in the brain parenchyma, as we previously found in 5XFAD mice following systemic immunomodulation[26,30]. Immunohistochemistry revealed that the infiltrating cells also expressed the anti-inflammatory cytokine, IL-10. Single-cell RNA-Seq revealed that the infiltrating monocyte-derived macrophages found within the brain expressed molecules that were not expressed by the resident resting or activated microglia,

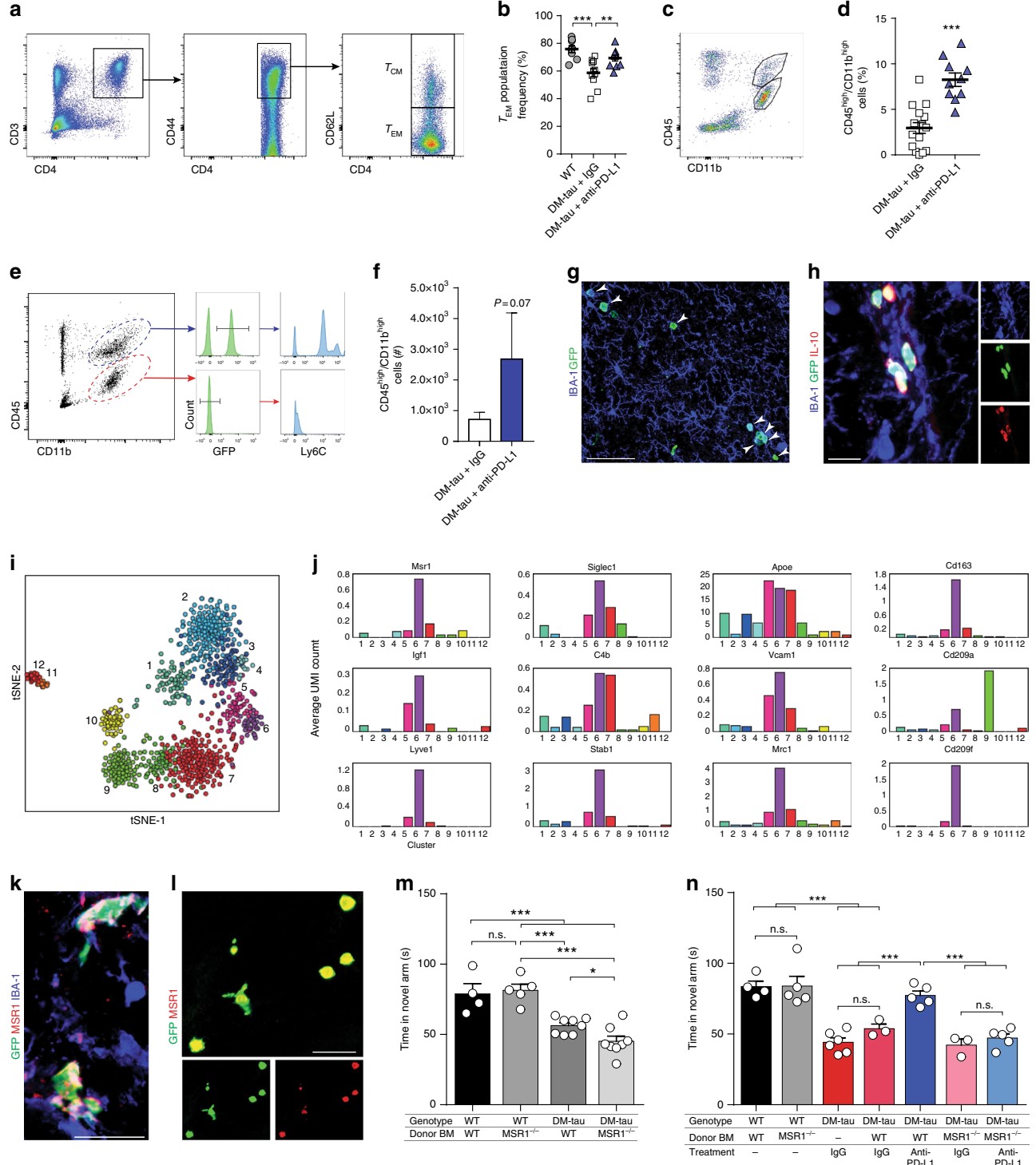

among which is the macrophage scavenger receptor, MSR1. Moreover, the potential of anti-PD-L1 to modify the disease was abrogated in chimeric DM-hTAU mice reconstituted with MSR1-deficient BM, suggesting that the effect of the monocyte-derived macrophages, is at least in part, via their MSR1 activity. Overall, the phenotype of the infiltrating monocyte-derived macrophages observed here is reminiscent of the phenotype of the cells previously shown to enter the CNS parenchyma through the CP, which promote the resolution of inflammation and contribute to spinal cord repair[52].

The involvement of local brain inflammation in AD pathogenesis[55], with the failure of systemic immunosuppressive drugs

to modify the disease in patients[55,56], along with our previous studies showing that transient reduction of systemic FoxP3 regulatory T cells is needed for homing of regulatory cells to the diseased brain[26], emphasize the need to distinguish between peripheral and local contributions of immunoregulatory cytokines, such as IL-10, in brain pathologies. For example, APP/PS1 AD-Tg mice with IL-10 deficiency were found to display reduced cerebral Aβ plaque load and reduced synaptic and cognitive deficits[27], while chronic overexpression of IL-10 in TgCRND8 AD-Tg mice[57], which exhibit IL-10 elevation in both the CNS and plasma, exacerbated disease pathology. In the present study, the beneficial effect of transiently boosting systemic immunity

**Fig. 5** Monocyte-derived macrophages uniquely affect disease modification in PD-L1 blockade in DM-hTAU mice. **a, b** Flow cytometry of splenocytes, CD44+CD62L$^{low}$ effector memory T (T$_{EM}$) cells, versus CD44+CD62L$^{high}$ central memory T (T$_{CM}$) cells in DM-hTAU mice, treated with 0.5 mg of anti-PD-L1 ($n = 10$) or IgG ($n = 11$) (one-way ANOVA, Fisher's exact test). **c, d** Flow cytometry of brains from anti-PD-L1-treated mice ($n = 10$), and IgG-treated mice ($n = 16$) analyzed for CD45$^{high}$CD11b$^{high}$, pooled from two experiments. **e–g** Repeated experiment as in **a, b** using GFP-BM-chimeric DM-hTAU mice. **e** Flow cytometry of GFP-labeled cells gated from CD45$^{high}$CD11b$^{high}$ cells, expressing Ly6C. **f** Quantitation of the number of GFP+CD45$^{high}$CD11b$^{high}$ cells in anti-PD-L1 ($n = 4$), relative to IgG-treated mice ($n = 6$). **g** Representative projections of confocal z-axis stacks, showing colocalization of GFP+ cells (green) with IBA-1 (blue), in the cortex of DM-hTAU$^{GFP/+}$ mice, treated with anti-PD-L1 antibody (arrowheads). Scale bar: 100 μm. **h** Representative confocal z-axis stacks, showing colocalization of GFP+ cells (green), IBA-1 (blue), and IL-10 (red) in the brains of anti-PD-L1-treated DM-hTAU$^{GFP/+}$ mice. Scale bar: 50 μm. **i** Sorted CD45$^{high}$CD11b$^{high}$ from DM-hTAU mice treated with anti-PD-L1, analyzed by single-cell RNASeq (Supplementary Fig. 8). tSNE plot depicting 899 cells. Clusters indicated by color and number. **j** Average Unique Molecular Identifier counts for selected genes across the 12 clusters. **k, l** Representative projections of confocal z-axis stacks, showing colocalization of GFP+ cells (green) with MSR1 (red) and IBA-1 (blue) in the cortex (**k**), and of GFP+ cells (green) with MSR1 (red) in the hippocampus of DM-hTAU$^{GFP/+}$ mice treated with anti-PD-L1 antibody (**l**). Scale bars: 25 and 50 μm. **m, n** BM-chimeric DM-hTAU and WT mice (male and female) prepared using WT or MSR1$^{-/-}$ mice as BM donors. **m** T-maze task, 2 weeks after BM transplant, of WT > WT ($n = 4$), MSR1$^{-/-}$ > WT ($n = 5$), WT > DM-hTAU ($n = 8$) and MSR1$^{-/-}$ > DM-hTAU ($n = 8$) chimeric mice. **n** The same mice were treated after the behavioral assessment in **m** with 1.5 mg of anti-PD-L1 antibody or IgG control antibody, and were tested again 1 month later for their performance in T-maze; nonchimeric IgG-treated DM-hTAU littermates were used as additional controls. Improved performance of WT > DM-hTAU treated with anti-PD-L1 ($n = 5$) versus IgG-treated WT > DM-hTAU ($n = 3$) and IgG-treated nonchimeric DM-hTAU mice ($n = 6$). MSR1$^{-/-}$ > DM-hTAU mice failed to show beneficial effect following treatment with anti-PD-L1 ($n = 5$), performing similarly to MSR1$^{-/-}$ > DM-hTAU treated with IgG ($n = 3$). In all panels, error bars represent mean ± s.e.m.; *$P < 0.05$, **$P < 0.01$, ***$P < 0.001$ (one-way ANOVA and Fisher's exact test)

that facilitated recruitment of immunomodulatory cells to the brain, is in line with our contention that this delicate balance requires optimizing immunity within the brain and in the circulation.

Several independent studies have shown that infiltrating monocyte-derived macrophages and T cells have beneficial effects within the brain in modulating the neuroinflammatory milieu, in general, and mitigating AD, in particular[21,22,52,58,59]. Thus, for example, in models of acute CNS insults, infiltrating monocyte-derived macrophages were shown to contribute to resolution of local inflammation and scar degradation[49,52,60,61]. In the context of neurodegenerative conditions, the increased abundance of monocyte-derived macrophages in the diseased brain was linked to removal of amyloid plaques, and reduced gliosis[21–23,59,62]. T cells, including regulatory T cells, could also infiltrate the CNS as a result of the systemic boosting of effector memory T cells, and could locally contribute to resolution of pathology[26,60]. Overall, the complex brain-immune relationships, in which the immune profile of the blood does not precisely mirror that of the brain in these diseases[28], could explain why numerous clinical trials, using anti-inflammatory drugs in AD have failed[9,55].

Brain inflammation was shown to have a negative impact on different aspects of brain plasticity, including cognition[41,63–67] and neurogenesis[68]. In the context of dementia and AD models, the local inflammatory milieu was suggested to accelerate tau phosphorylation via the activation of kinases[46,69–71], and to enhance plaque burden[55,63,64]. In addition, the complement-dependent pathway in microglia, which is involved in pruning excess synapses in development, has been suggested as a negative player in AD[34]. Here, we show that IL-1β was elevated in DM-hTAU, and was reduced as a result of the treatment with anti-PD-L1. Increased levels of IL-1β have been observed in patients with AD and vascular dementia[72]. Moreover, it was reported in mouse models of AD that inhibition of IL-1β signaling reduces the activity of several tau kinases in the brain, and also reduces phosphorylated tau levels[43]. The results presented here suggest that astrocytes are the predominant cells that express this inflammatory cytokine. Importantly, the observed reduction of IL-1β as a result of the PD-L1 blockade treatment in the present study could reflect a direct effect of the infiltrating myeloid cells on astrocytes, or an indirect mechanism via their effect on microglia[73], as it was previously demonstrated that microglia can modulate the inflammatory response in astrocytes[74].

Our present finding that treatment with immune checkpoint blockade is beneficial in both the beta-amyloid model of AD as well as in the DM-hTAU model of dementia supports our contention that the treatment activates a universal immune repair mechanism. This reparative mechanism, which starts outside the brain and culminates in a sequence of events within the brain, does not directly target any of the brain pathologies that are hallmarks of the disease. Therefore, testing any immune checkpoint antibody in models of AD should be based first on the ability of the antibody to activate systemic immunity, and subsequently to improve cognitive performance, and only then be tested for its ability to reduce disease pathology[26,30]; measuring only plaque accumulation cannot serve as a primary read out, as the effect of anti-PD1/PDL-1 antibodies on the plaque load is secondary. The need to optimize the treatment with the tested antibody in each animal model of AD at each facility might explain the failure of a study using PD-1 blockade, that measured only plaque load[75].

The two mouse models that we used here represent different forms of inherited AD and dementia[29,31]. Since these transgenic models are neuronal driven our results strongly support our approach of activating the immune system to revitalize a common pathway of repair that is independent of the manner in which the disease pathology was caused or induced. In light of these findings, another study using PD-1 blockade in a murine Toxoplasma infection model demonstrated diminished cyst burden due to leukocytes infiltration via the choroid plexus[76]. Taken together, these results further substantiate the effect of immune checkpoint blockade in augmenting systemic immunity, to facilitate the mobilization of leukocytes to sites of neuroinflammation, leading to immune modulation and tissue repair[77,78].

In the experiments described here, a single injection of the antibody led to reversal of cognitive deficits, assessed 1 month after treatment. When repeated injections were given, the antibodies were administered once a month. Given the short half-life of the rat anti-mouse PD-L1 antibody that was used here, it is suggested that the beneficial effect of the immunotherapy for AD and dementia does not require continuous exposure to the antibody, and that the effect is mechanistically different from that underlying the current anti-PD-L1 treatment used in cancer therapy[79,80]; it is therefore expected to have minimal adverse immunological effects. Further studies are currently being performed to optimize the treatment regimen for translation of this therapy to AD patients.

In conclusion, immune checkpoint blockade is the first proposed treatment for AD that targets common pathways in the

immune system rather than any single disease-escalating factor within the brain, aiming to revive the body's own immune-based repair mechanism that could comprehensively modify the diseased brain, irrespective of the primary disease etiology.

## Methods

**Animals**. Heterozygous 5XFAD transgenic mice (Tg6799; on a C57/BL6-SJL background) co-overexpress mutant forms of human APP associated with familial AD, the Swedish mutation (K670N/M671L), the Florida mutation, (I716V), and the London mutation (V717I). Heterozygous DM-hTAU transgenic mice expressing two mutations K257T/P301S (double mutant, DM; on a BALBc-C57/BL6 background) under the natural TAU promoter, associated with severe disease manifestations of frontotemporal-dementia in humans, were kindly provided by Prof. Dan Frenkel. Genotyping was performed by PCR analysis of tail DNA, as previously described[31]. Throughout the study, WT controls in each experiment were nontransgene littermates from the relevant tested mouse colonies. C57BL/6 CD45.2 Ub-GFP mice, in which GFP is ubiquitously expressed[48], were used as donors for bone marrow chimeras. MSR1[−/−] were generated by Professor Tatsuhiko Kodama, and were kindly provided by Prof. Dan Frenkel[51]. Animals were bred and maintained by the Animal Breeding Center of the Weizmann Institute of Science. All experiments detailed herein complied with the regulations formulated by the Institutional Animal Care and Use Committee (IACUC) of the Weizmann Institute of Science.

**RNA purification, cDNA synthesis, and quantitative real-time PCR analysis**. Total RNA of the hippocampal DG was extracted with the TRI Reagent (Molecular Research Center) and purified from the lysates using an RNeasy Kit (Qiagen). The expression of specific mRNAs was assayed using fluorescence-based quantitative real-time PCR (RT-qPCR). RT-qPCR reactions were performed using Fast-SYBR PCR Master Mix (Applied Biosystems). Quantification reactions were performed in triplicate for each sample using the standard curve method. Peptidylprolyl isomerase A (ppia) was chosen as a reference (housekeeping) gene. The amplification cycles were 95 °C for 5 s, 60 °C for 20 s, and 72 °C for 15 s. At the end of the assay, a melting curve was constructed to evaluate the specificity of the reaction. All RT-qPCR reactions were performed and analyzed using StepOne software V2.2.2 (Applied Biosystems).

The following primers were used for the genes indicated:

ppia forward 5′-AGCATACAGGTCCTGGCATCTTGT-3′ and reverse 5′-CAAAGACCACATGCTTGCCATCCA-3′;

tnf-a forward 5′-GCCTCTTCTCATTCCTGCTT-3′ reverse CTCCTCCACTTGGTGGTTTG-3′;

il-12p40 forward 5′-GAAGTTCAACATCAAGAGCA-3′ and reverse 5′-CATAGTCCCTTTGGTCCAG-3′;

il-10 forward 5′-TGAATTCCCTGGGTGAGAAGCTGA-3′ and reverse 5′-TGGCCTTGTAGACACCTTGGTCTT-3′;

il-6 forward 5′-AACAAGAAAGACAAAGCCAG-3′ and reverse 5′-GGAGAGCATTGGAAATTGG -3′;

il-1β forward 5′-CCAAAAGATGAAGGGCTGCTT-3′ and reverse 5′-TGCTGCTGCGAGATTTGAAG -3′;

**Immunohistochemistry**. Mice were transcardially perfused with phosphate buffered saline (PBS) before tissue excision and fixation. Tissues that were not adequately perfused were not further analyzed, to eliminate autofluorescence associated with blood contamination. Two different tissue preparation protocols (paraffin-embedded or microtome free-floating sections) were applied, as previously described[25,26]. The following primary antibodies were used: mouse anti-Aβ (1:300, Covance, #SIG-39320); rabbit anti-GFAP (1:200, Dako, #Z0334); chicken anti-GFAP (1:400, Abcam, #4674); rabbit anti-cleaved caspase 3 (1:100, Cell Signaling, #9664); mouse anti-AT-100 and AT-180 (1:50, Innogenetics, #90209 and #90337); mouse anti-Neu-N (1:100, Millipore, # MAB377); rabbit anti-synaptophysin (1:100, Abcam, #32127); mouse anti-Vglut1 (1:100, Millipore, MAB5502); goat anti-IBA-1 (1:100, Abcam #5076); rabbit anti-IBA-1 (1:200, Wako #019-19741); mouse anti-IBA-1 (1:100, GeneTex, #GTX632426); rabbit anti-GFP (1:100, MBL, #598); goat anti-GFP (1:100, Abcam, #6658); rabbit anti-IL1β (1:100, Santa Cruz Biotechnology, SC-7884); goat anti-IL-10 (1:50, R&D Systems, Minneapolis, MN, #AF519); rabbit anti-MSR1 (1:100, GeneTex, #GTX51749). Secondary antibodies included: Cy2/Cy3/Cy5-conjugated donkey anti-mouse/goat/rabbit/rat antibodies (1:200; all from Jackson Immunoresearch). The slides were exposed to 4′,6-diamidino-2-phenylindole (DAPI) for nuclear staining (1:10,000; Biolegend) for 1 min. Two negative controls were routinely used in immunostaining procedures, staining with isotype control antibody followed by secondary antibody, or staining with secondary antibody alone. The tissues were applied to slides, mounted with Immu-mount (9990402, from Thermo Scientific), and sealed with cover-slips. Microscopic analysis was performed using a fluorescence microscope (E800; Nikon) or laser-scanning confocal microscope (Zeiss, LSM880). The fluorescence microscope was equipped with a digital camera (DXM 1200F; Nikon), and with either a ×20 NA 0.50 or ×40 NA 0.75 objective lens (Plan Fluor; Nikon). Recordings were made on postfixed tissues using acquisition software (NIS-Elements, F3 (Nikon) or LSM (Carl Zeiss, Inc.)). For quantification of staining

intensity, total cell and background staining were measured using ImageJ software (NIH), and the intensity of specific staining was calculated, as previously described[81]. Images were cropped, merged, and optimized using Photoshop CS6 13.0 (Adobe), and were arranged using Illustrator CS5 15.1 (Adobe).

**Radial arm water maze (RAWM)**. The RAWM task was used to test spatial learning and memory, as was previously described in detail[82]. Briefly, six stainless-steel inserts were placed in the tank, forming six swim arms radiating from an open central area. The escape platform was located at the end of one arm (the goal arm), 1.5 cm below the water surface, in a pool 1.1 m in diameter. The water temperature was maintained between 21 and 22 °C. Water was made opaque with milk powder. Within the testing room, only distal visual shape and object cues were available to the mice to aid in location of the submerged platform. The goal arm location remained constant for a given mouse. On day 1, mice were trained for 15 trials (spaced over 3 h), with trials alternating between a visible and hidden platform, and the last 4 trails with hidden platform only. On day 2, mice were trained for 15 trials with the hidden platform. Entry into an incorrect arm, or failure to select an arm within 15 s, was scored as an error. Spatial learning and memory were measured by counting the number of arm entry errors or the escape latency of the mouse on each trial. Training data were analyzed as the mean errors or escape latency, for training blocks of three consecutive trials. The investigator was blind to the identity of the animals throughout the experiments. Data were analyzed and codes were opened by a member of the team who did not perform the behavioral tests.

**T-maze**. The T-maze test assesses spatial short-term memory and alternation behavior, analyzing the animals' ability to recognize and differentiate between a novel unknown versus a familiar compartment[31,40]. The T-shaped maze was made of plastic with two 45 cm long arms, which extended at right-angles from a 57 cm long alley. The arms had a width of 10 cm and were surrounded by 10 cm high walls. The test consisted of two trials at an interval of 5 min, during which time the animals were returned to their home cages. During an 8 min acquisition trial, one of the short arms was closed. In the 3 min retention trial, mice had access to both arms and to the alley. Time spent in each of the arms and in the long alley was assessed. Cognitively healthy mice tend to spend more time in the novel arm than in the familiar one or in the alley. Data were recorded using the EthoVision XT 11 automated tracking system (Noldus Information Technology). The investigator was blind to the identity of the animals throughout the experiments. Data were analyzed and codes were opened by a member of the team who did not perform the behavioral tests.

**Y-maze**. Spontaneous alternation behavior was recorded in a Y-maze to assess short-term memory performance[39]. The apparatus was a symmetrical Y-maze; each arm measured 50 × 10 cm, with 40-cm-high side walls. Mice were placed in the maze and allowed to freely explore for 5 min. Arms were arbitrarily labeled A, B, and C, and the sequence of arm entries was used to assess alternation behavior. An alternation was defined as consecutive entries into all three arms. The number of maximum alternations was therefore the total number of arm entries minus two, and the percentage of alternations was calculated as (actual alternations/maximum alternations) × 100. For example, if arms referred to as A, B, C, if the mouse performed ABCABCABBAB, the number of arm entries would be 11, and the successive alternations: ABC, BCA, CAB, ABC, BCA, CAB. Therefore, the percentage of alternations would be $[6/(11-2)] \times 100 = 66.7$[83]. Statistical analysis was performed using analysis of variance (ANOVA) and the Fisher's exact test.

**Bone marrow chimerism**. BM chimeras were prepared as previously described[49]. In brief, chimeras were prepared by subjecting gender-matched recipient mice to lethal irradiation (950 rad), directing the beam to the lower part of the body, and avoiding the head. The mice were then reconstituted with 5×10[6] GFP-BM cells. The mice were analyzed 5 weeks after BM transplantation (exhibiting an average of 72% chimerism). CNS-infiltrating GFP[+] myeloid cells were verified to be CD45[high]/CD11b[high], representing monocyte-derived macrophages and not microglia[52]. In order to study the role of MSR1[+] monocytes-derived macrophages, gender-matched DM-hTAU and WT littermates were subjected to whole body irradiation (950 rad). The mice were then reconstituted either with 5×10[6] MSR[−/−]-BM cells or with WT BM cells, derived from non-transgene age-matched littermates.

**Cresyl Violet staining**. Fixed brains were sagittally sectioned, with a section thickness of 6 μm. To estimate neuronal survival, Cresyl violet staining was performed to visualize neurons. Pyramidal neurons were counted in each brain from serial sections located 30 μm apart. All cell counting were performed by a researcher who was blind to the identity of the animals.

**Therapeutic antibodies**. For PD-1 blockade, PD-1-specific blocking antibody (anti-PD-1; rat IgG2a isotype; clone RPM1-14; BIOXCELL) and isotype control (anti-trinitrophenol; clone 2A3, BIOXCELL) were administered i.p. For PD-L1 blockade, throughout the entire study, PD-L1-blocking antibody directed to mouse PD-L1 was used (anti-PD-L1 antibody; rat IgG2b isotype; clone 10F.9G2; BIOXCELL) and isotype control (anti-keyhole limpet hemocyanin; clone LTF-2;

BIOXCELL) was administered i.p. In one experiment, anti-human PD-L1 antibody was used, which was produced as follows: The V-gene sequences of the anti-mouse anti-PD-L1 antibody YW243-55-S70 (Ref US20100203056A1) was synthesized and cloned onto coding regions for murine IgG2a/VL-K constant domains. Subsequently, the antibody transiently expressed in HEK 293 cells and purified standard protocols[44].

**Aβ plaque quantitation**. From each brain, 6 μm coronal slices were collected, and five sections per mouse were immunostained, from 4 to 5 different predetermined depths throughout the region of interest (DG or cerebral cortex). Histogram-based segmentation of positively stained pixels was performed using Image-Pro Plus software (Media Cybernetics, Bethesda, MD, USA). The segmentation algorithm was manually applied to each image, in the DG area or in cortical layer V, and the percentage of the area occupied by total Aβ immunostaining was determined. Plaque numbers were quantified from the same 6 μm coronal brain slices, and are presented as average number of plaques per brain region, in the region of interest (ROI), identically marked on all slides from all depths and in all animals examined. Prior to quantification, slices were coded to mask the identity of the experimental groups, and were quantified by an observer blinded to the identity of the groups.

**Aggregated tau quantitation**. After perfusion, hippocampus and cortex tissues were dissected and homogenized in ice-cold buffer A (349.1 mM sucrose, 0.1 mM $CaCl_2$, 1 mM $MgCl_2$) supplemented with protease inhibitor cocktail (Sigma; P8340). Homogenates were diluted in TBS with 1% Triton X-100 (Sigma; T8787) supplemented with protease inhibitor cocktail, and were individually measured by Tau Aggregation Assay using a commercially available kit (CisBio; CB-6FTAU-PEG) according to the manufacturer's instructions. This assay is based on the fluorescence resonance energy transfer (FRET) immunoassay. Protein concentrations were measured using BCA protein assay kit (Pierce; 23227) according to the manufacturer's instructions.

**IL-1β quantitation**. Hippocampal tissue homogenates in buffer A supplemented with protease inhibitor cocktail (as described above) were measured by Mouse IL1 beta assay using a commercially available kit (CisBio; CB-62MIL1BPEG) according to the manufacturer's instructions, and normalized to protein concentration. This assay as above, was based on the FRET immunoassay.

**Flow cytometry sample preparation and analysis**. Mice were transcardially perfused with PBS, and tissues were treated as previously described[26]. Brains were dissociated using the gentleMACS dissociator (Miltenyi Biotec). Spleens were mashed with the plunger of a syringe and treated with ACK (ammonium chloride potassium)-lysing buffer to remove erythrocytes. In all cases, samples were stained according to the manufacturers' protocols. All samples were filtered through a 70-μm nylon mesh, and blocked with anti-Fc CD16/32 (1:100; BD Biosciences). The following fluorochrome-labeled monoclonal antibodies were purchased from BD Pharmingen, BioLegend, R&D Systems or eBiosciences, and used according to the manufacturers' protocols: Brilliant-violet 421-conjugated anti-CD45 or CD-4; PE-conjugated anti-CD3 or anti-CD11b; FITC-conjugated anti-CD44 or anti-CD11b; PerCP-Cy5.5-conjugated anti-CD62L; APC-conjugated anti-Ly6C. Cells were analyzed on an LSRII cytometer (BD Biosciences) using FlowJo software. In each experiment, relevant negative control groups, positive controls and single-stained samples for each tissue were used to identify the populations of interest and to exclude other populations.

**Sorting of myeloid cells**. Cell populations were sorted with FACSAriaIII (BD Biosciences, San Jose, CA). Prior to sorting, all samples were filtered through a 40-μm nylon mesh. For the isolation of monocytes-derived macrophages, samples were gated for CD45[high] and CD11b[high] (Brilliant-violet 421, 1:150, 30-F11, Biolegend Inc. San Diego, CA; APC CD11b, 1:100, M1/70, eBioscience), while excluding doublets. Isolated cells were single cell sorted into 384-well cell capture plates containing 2 μL of lysis solution and barcoded poly(T) reverse-transcription (RT) primers for single-cell RNA-seq[84]. Four empty wells were designated in each 384-well plate as a no-cell control during data analysis. Immediately after sorting, each plate was spun down to ensure cell immersion into the lysis solution, snap frozen on dry ice, and stored at −80 °C until processing.

**Preparation of massively parallel single-cell RNA-seq library (MARS-seq)**. Single-cell libraries were prepared as previously described[73]. In brief, mRNA from cells sorted into cell capture plates was barcoded, converted into cDNA, and pooled using an automated pipeline. The pooled sample was then linearly amplified by T7 in vitro transcription, and the resulting RNA was fragmented and converted into a sequencing-ready library by tagging the samples with pooled barcodes and Illumina sequences during ligation, RT, and PCR. Each pool of cells was tested for library quality, and concentration was assessed, as described[73].

**Analysis of single cell RNA-seq data**. All MARS-seq libraries were sequenced using an Illumina NextSeq 500 at an average sequencing depth of 50,000 reads per cell. Sequences were demultiplexed, mapped and filtered as previously described[84],

extracting a set of unique molecular identifiers per cell. Cells were then clustered using the MetaCell analysis package[85]. Briefly, informative genes were used to compute cell-to-cell similarity and to build a K-nn graph ($k = 50$) to group cells into cohesive groups (or "meta-cells"). Finally, the package uses bootstrapping to derive strongly separated clusters. The MetaCell package is available at https://bitbucket.org/tanaylab/metacell/src

**Statistical analysis**. The specific tests used to analyze each set of experiments are indicated in the figure legends. Data were analyzed using a two-tailed Student's $t$ test to compare between two groups; one-way ANOVA was used to compare several groups, followed by the Fisher's exact test post-hoc procedure. Data from behavioral tests were analyzed using two-way repeated-measures ANOVA, and Dunnetts' post-hoc procedure was used for multiple comparisons. Sample sizes for behavioral studies were chosen with adequate statistical power based on the literature and past experience, and mice were allocated to experimental groups according to age, gender and genotype. Investigators were blinded to the identity of the groups during experiments and outcome assessment. All inclusion and exclusion criteria were pre-established according to IACUC guidelines. Results are presented as mean ± s.e.m. In the graphs, $y$-axis error bars represent s.e.m. Statistical calculations were performed using GraphPad Prism software (GraphPad Software, San Diego, CA).

## Data availability

The authors declare that the data supporting the findings of this study are available within the paper and its Supplementary Information files or from the corresponding author on reasonable request.

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

## Acknowledgements

We thank Prof. Dan Frenkel (Tel-Aviv University, Israel) for his generous gift of DM-hTAU and MSR1$^{-/-}$ mice and Dr. Shelley Schwarzbaum for editing the manuscript. Research in the M.S. lab is supported by Advanced European Research Council grants (232835), and by the EU Seventh Framework Program HEALTH-2011 (279017); Israel Science Foundation (ISF)-research grant no. 991/16; and ISF-Legacy Heritage Biomedical Science Partnership-research grant no. 1354/15. We wish to thank the Adelis Foundation for their generous support of our AD research. M.S. holds the Maurice and Ilse Katz Professorial Chair in Neuroimmunology. All listed authors were either students or employees at the Weizmann Institute of Science while conducting their work.

## Author contributions

M.S. supervised the entire study. N.R. designed the experiments under the supervision of M.S. K.B. was involved in the design of part of the experiments. N.R. performed the experiments with the assistance of A.T.-K., H.B.-Y., P.W.-R., L.C., A.K., K.B. (under the supervision of M.S.). R.D.-S., H.K.-S. and A.W. performed single-cell RNA-seq under the supervision of M.S and I.A. The manuscript was written by M.S. and N.R.

## Additional information

**Competing interests:** M.S., N.R. and K.B. are inventors of intellectual property related to this study. M.S. serves as a consultant of ImmunoBrain Checkpoint Ltd. The remaining authors declare no competing interests.

