## [Peer Review File · Nature Communications]

Reviewers' comments:

Reviewer #1 (Remarks to the Author):

In this manuscript, Rosenzweig et al. claim that blocking immune checkpoints is effective to symptomatic and histopathological phenotypes of tauopathy in the DM-hTAU mouse model. In a previous paper, the authors showed that activating systemic immunity using an antibody against PD-1 reversed cognitive deficits and histopathological changes of the APP-transgenic 5XFAD mouse model. In this study, they found that treatment with anti-PD-1 antibody is effective against the phenotypes of 5XFAD female mice and that anti-PD-L1 antibody is similarly effective. In addition, the authors found that blocking functions of either PD-1 or PD-L1 is effective in mitigating cognitive decline and histopathological findings of DM-hTAU mice, an animal model of frontotemporal dementia (FTD). These treatment was associated with increased monocyte-derived macrophages within the brain parenchyma and reduction in tau hyperphosphorylation. This study is potentially of interest but seems to remain descriptive. This reviewer has some concerns as follows.

Major points;

(1) This manuscript, especially a part showing the effect of PD-1/PD-L1 axis blockade on tauopathy, seems descriptive. The authors have not addressed the mechanism underlying the effect of PD-1/PD-L1 blockade on primary tauopathy (tau gene mutation-induced tauopathy). It is critical for this study how PD-1/PD-L1 blockade affects intracellular tau pathology in the absence of extracellular amyloid pathology. The authors cite 3 papers and state that "neuroinflammation in animal models of tauopathies was shown to enhance tau hyperphosphorylation, which accelerates disease progression" (page 8, lines 220-221). But, two of cited papers are review papers and the remaining one (Ghosh S, et al. J Neurosci 33:5053-64, 2013) showed that sustained interleukin-1beta overexpression exacerbates tau pathology of an AD model mice, which were associated with extracellular amyloid pathology. The authors show that PD-1/PD-L1 blockade enhanced recruitment of monocyte-derived macrophages to the brain parenchyma, but they have not addressed a question whether increased macrophages mediates improvement of tau phosphorylation and cognitive decline of DM-hTAU mice.

(2) The authors used 5XFAD transgenic mice co-overexpressing human mutant APP and Presenilin-1 and DM-hTAU transgenic mice overexpressing K257T/P301S-double mutant tau. Artificial overexpression of these transgenes may add unexpected outcomes to the pathogenic process of AD or FTD. Using the other models such as gene knockin-based model mice, the authors should rule out the possibility that PD-1/PD-L1 blockade only mitigates artifacts caused by Tg overexpression.

(3) It is not clear whether PD-1/PD-L1 blockade solely suppressed occurrence of brain pathology in model mice or also reversed the pathology by cleaning up deposited aggregates.

(4) Disease modifying therapies for neurodegenerative diseases such as AD and FTD require a more than decade-long treatment. Can the authors expect no adverse effect after long-term blockade of PD-1/PD-L1 axis in clinical usage?

Minor points;

(1) Did PD-1/PD-L1 blockade suppress neuronal loss of DM-hTAU mice?

(2) Fig 1: Pictures of the wild-type control should be shown in (e) and (f). No explanation of arrowheads in (e). No legend for (h).

(3) Figs 2 and 4: Expression levels of synaptophysin and phosphorylated tau were evaluated using immunohistochemistry, which is not suitable for quantitative analysis. They should apply another

quantitative method such as immunoblotting.

(4) Page 7, lines 193-195: DM-hTAU transgenic mouse is a model of FTD but not of AD.

(5) Typos;

Page 3, line 75; preedisposition

Page 17, line 486; 5 weeks weeks after ...

Reviewer #2 (Remarks to the Author):

This study by Rosenzweig.N et al. aim to determine whether PD-1/PD-L1 immune checkpoint blockade could reduce cognitive loss and pathology in a mouse model of tau associated dementia, which further prove the potential application of PD-1/PD-L1 treatment for AD and other neurodegenerative diseases. In general, the experiments are convincing in term of the experiments were well-performed and controlled. However, there are several conceptual and experimental concerns detailed below.

Major points:

1. In figure 1e and 1f, besides neuronal loss showed in the subiculum, images of the cortex are also needed to comprehensively demonstrate that PD-1 blockade treatment could reduce neuronal loss in the brains. Furthermore, it would be more convincing to include WT control mice since neuronal survival in 5XFAD mice were calculated relative to WT mice. For figure 1g, the authors need to provide larger field of images vision.
2. In figure 2j, the authors only showed the changes of il-12p40/ il-10 ratio. It is more accurate to show the relative expression level of il-12p40 and il-10 separately. In addition, the authors should measure the expression of other pro-inflammatory cytokines, i.e. tnf- α and il-6. The reduction of these proinflammatory cytokines would be better evidence for modulation of neuroinflammation response by PD-1/PD-L1 blockade in these mice.
3. As to RAWM experiments in figure 1 and figure 2, it is not clear how the authors determined a mouse with motor deficit and how many mice had motor deficits. Are there any differences between treated and untreated mice? It is necessary to provide additional data to demonstrate that the motor ability of mice analyzed in RAWM experiments is same among different groups.
4. In figure 3, it is not clear what ages of animals were used in the T maze and Y maze behavioral tests? Since the spatial memory ability of both female and male mice were measured using these two behavior test, it is curious about how the female mice behave in Y maze, which was not showed by the authors.
5. For figure 4e, the authors have previously demonstrated that increased dose of PD-L1 blockade had little effect on enhancing cognitive performance in 5XFAD mice, showed in figure 2b. So it is not significant and might be removed to supplementary data. Additionally, according to similar results, it seems that antibody efficacy is unlikely the dominant factor for treatment in different animal models, which discussed by the authors in line 344-352.
6. In figure 5a, effector memory T cells (CD44+CD62Llow) are abbreviated as "TEM", but not "TEF". Are these representative figures from PD-L1 treated mice? It would be more convincing to include WT and IgG isotype control groups. It would be also important to include absolute cell count of lymphocyte populations in Figure 5b and 5d, since cell frequency cannot accurately reflect changes in

the cell numbers. In the BM chimeric experiment, it is not clear whether part of CD45+CD11b- cells are GFP+, which indicate the infiltrating lymphocytes (including T cells or B cells). If so, further phenotypic and functional analysis of this cell subset and monocyte-derived macrophages would probably reflect how systemic PD-1/PD-L1 blockade affect neuroinflammation and pathological process in the brain.

7. Although the results shown in this manuscript and recent work (Nature medicine, 2016) are compelling, a significant conceptual concern from this reviewer is the difference between the findings of this group compared to the findings of Latta-Mahieu, M. et al. (Glia, 2018). Different antibody efficacies discussed by the authors in the text might not be the only reason account for the contradictory results. At least in part, CNS inflammation at preclinical and clinical AD stage could serve different roles in disease progression. To move the field forward, it is important to either perform additional experiments and/or discuss the potential reasons in depth.

Minor points:

1. In the text and figure1: should be "NeuN" but not "Neu-N".
2. Line 151-152: the description of PD-L1 expression is not correct for its expression in a wide variety of cell types.
3. Line 193 should be rephrased. Transgenic mice with FTD-associated tau mutations are not considered as AD mouse models.
4. Line 258: should be "CD62L" but not "CD62-L".
5. Line 358-359: Please explain why antibodies were administered once a month? Why not a longer or shorter interval?
6. Line 416: missing closing parenthesis after text "#SIG-39320".
7. Line 867: effector memory T cells should be abbreviated as "TEM"

Reviewer #3 (Remarks to the Author):

Based on observations that the recruitment of monocyte-derived macrophages to the brain is dependent on systemic availability of IFN γ -producing CD4+ve T cells, the authors speculated that increasing the number of IFN γ -positive cells systemically or at the level of the choroid plexus would promote the recruitment of monocyte-derived macrophages to modulate brain inflammation. They previously demonstrated the benefits of this approach in neurodegenerative disease models by reducing the numbers of FoxP3 regulatory T cells or by blocking the PD-1 immune checkpoint in the 5xFAD mouse model of amyloid pathology. They extend this work here to show that targeting PD-1 or PD-L1 ligands with different treatment regimes shows benefit in 5xFAD mice and in a human tauopathy mouse model. In the tau mice, administration of an anti-PD-L1 antibody increased monocyte-derived macrophages in the brain, mediated local inflammatory responses and reduced tau phosphorylation and cognitive dysfunction. This is an interesting study describing a new approach to ameliorate neurodegenerative disease by modulating systemic inflammation. The novelty of the work is a little reduced because of previous work in this area by the authors, but this paper does show that the approach has utility in distinct models of human neurodegeneration linked to Alzheimer's disease.

The writing and data are high quality, the work is likely to still be of interest to the field, and methods used and statistical analysis performed appear appropriate.

Comments:

1. Nature journals recommend following the ARRIVE reporting guidelines when documenting animal studies (PLoS Bio 8(6), e1000412,2010). This includes reporting the effects of treatment/modifications in wild-type in addition to experimental animals. These controls are missing from the current manuscript which means that it is not possible to ascertain if the effects of treatment are related to disease development in the transgenic mice. The background strain on which DM-htau mice are bred should also be stated.

2. Throughout the manuscript, the DM-htau mice are referred to as a mouse model of AD. It would be more accurate to call these a mouse model of tauopathy since the tau mutations expressed by these mice are not linked with AD (no tau mutations are) and as the author correctly states these mutations instead cause FTD.

3. Fig. 1E. Neuronal survival should be reported for the hippocampus in addition to the subiculum since this is a more meaningful measurement in terms of cognitive decline in the 5xFAD mice.

4. Line 144-145 states that pathological analysis was carried out on only animals without a motor impairment. Presumably animals with motor problems were also excluded from the behavioural analysis? This should be more clearly stated. Related to this, why were only 6 WT animals assessed by cresyl violet? (line 774).

5. Fig. 2H-J. The conclusion that anti-PD-1/anti-PD-L1 protects synapses would be better demonstrated by immunolabelling with both pre- and post-synaptic markers to illustrate synaptic pairs that are likely to be functional and/or by visualisation and counting of synaptic spines in these groups of animals.

6. The analysis of tau changes in the DM-htau mice following treatment should be expanded to make these findings more compelling. I would prefer to see some biochemical analysis of changes in tau phosphorylation relative to total tau amounts for e.g. by immunoblotting. It would also be useful to examine which species of tau is affected, for example by extraction and quantification of sarkosyl-insoluble tau. It would be useful to explore the pathways that are implicated in these changes by either showing elevation in specific kinases or tau aggregate clearance mechanisms. In addition, the relationship between tau and synapse health should be examined, as described above for analysis of synapses in the 5xFAD mice.

7. Discussion: Please include some discussion about how anti-PD-1/anti-PD-L1 treatment is likely to affect astrocyte activation (Fig. 2d). Is it assumed that this results from lowered plaque burden?

8. Discussion: Please discuss recent papers from the AD field which have genetically modified IL-10 in APP transgenic mice (Chakrabarty et al., 2015 Neuron; Guillot-Sestier et al., 2015 Neuron) and that show effects of modulating this cytokine in terms of microglial responses and plaque burden distinct from the anti-inflammatory role of this cytokine. This is an important body of work to consider in light of the changes shown in this current manuscript.

typos

Line 106. Cohort

Below please find our point-by-point response to the referees' comments:

Reviewer #1 (Remarks to the Author):

In this manuscript, Rosenzweig et al. claim that blocking immune checkpoints is effective to symptomatic and histopathological phenotypes of tauopathy in the DM-hTAU mouse model. In a previous paper, the authors showed that activating systemic immunity using an antibody against PD-1 reversed cognitive deficits and histopathological changes of the APP-transgenic 5XFAD mouse model. In this study, they found that treatment with anti-PD-1 antibody is effective against the phenotypes of 5XFAD female mice and that anti-PD-L1 antibody is similarly effective. In addition, the authors found that blocking functions of either PD-1 or PD-L1 is effective in mitigating cognitive decline and histopathological findings of DM-hTAU mice, an animal model of frontotemporal dementia (FTD). This treatment was associated with increased monocyte-derived macrophages within the brain parenchyma and reduction in tau hyperphosphorylation. This study is potentially of interest but seems to remain descriptive. This reviewer has some concerns as follows.

Major points;

(1) This manuscript, especially a part showing the effect of PD-1/PD-L1 axis blockade on tauopathy, seems descriptive. The authors have not addressed the mechanism underlying the effect of PD-1/PD-L1 blockade on primary tauopathy (tau gene mutation-induced tauopathy). It is critical for this study how PD-1/PD-L1 blockade affects intracellular tau pathology in the absence of extracellular amyloid pathology. The authors cite 3 papers and state that "neuroinflammation in animal models of tauopathies was shown to enhance tau hyperphosphorylation, which accelerates disease progression" (page 8, lines 220-221). But, two of cited papers are review papers and the remaining one (Ghosh S, et al. J Neurosci 33:5053-64, 2013) showed that sustained interleukin-1beta overexpression exacerbates tau pathology of an AD model mice, which were associated with extracellular amyloid pathology. The authors show that PD-1/PD-L1 blockade enhanced recruitment of monocyte-derived macrophages to the brain parenchyma, but they have not addressed a question whether increased macrophages mediates improvement of tau phosphorylation and cognitive decline of DM-hTAU mice.

We agree with the referee's comments. We now provide additional mechanistic insight to the effect of the treatment in the tau model. We now show that the DM-hTAU mice exhibit an elevation in IL-1 β levels, the level of which was reduced following treatment (Fig. 5 h,j,l). Moreover, a correlation was found between the improved cognitive ability and the reduction in IL-1 β protein levels (Fig. 3m). In addition, in probing the mechanism for this response, we found, using chimeric DM-hTAU^{GFP/+} mice, that infiltrating monocyte-derived macrophages are the cells that express the anti-inflammatory cytokine, IL-10 (Fig. 5h), and that IL-1 β is expressed by astrocytes (Fig. 3k). Moreover, the infiltrating monocyte-derived macrophages cells were found mainly in close proximity to site of pathological tau (Fig. 5 j-l) and expressed a scavenger receptor that was not found to be expressed by microglia (Fig. 5i).

(2) The authors used 5XFAD transgenic mice co-overexpressing human mutant APP and

Presenilin-1 and DM-hTAU transgenic mice overexpressing K257T/P301S-double mutant tau. Artificial overexpression of these transgenes may add unexpected outcomes to the pathogenic process of AD or FTD. Using the other models such as gene knockin-based model mice, the authors should rule out the possibility that PD-1/PD-L1 blockade only mitigates artifacts caused by Tg overexpression.

This issue is now discussed (page17).

(3) It is not clear whether PD-1/PD-L1 blockade solely suppressed occurrence of brain pathology in model mice or also reversed the pathology by cleaning up deposited aggregates. Reverse?

We now provide new data demonstrating that the treatment, by behavioral criteria, was able to reverse and not only delay cognitive loss (Fig. 2m, n).

(4) Disease modifying therapies for neurodegenerative diseases such as AD and FTD require a more than decade-long treatment. Can the authors expect no adverse effect after long-term blockade of PD-1/PD-L1 axis in clinical usage?

The treatment was effective when a single injection was given, and cognitive assessment performed after one month. Moreover, monthly treatment was sufficient to maintain a long-term effect. We now discuss why we anticipate lower adverse effects of long-term treatment, based on the fact that unlike the case of cancer that requires continuous exposure to the antibody, in the present study intermittent treatment regimen is required (Pages 17-18).

Minor points;

(1) Did PD-1/PD-L1 blockade suppress neuronal loss of DM-hTAU mice?

We now added these results (Fig. 4f, g).

(2) Fig 1: Pictures of the wild-type control should be shown in (e) and (f). No explanation of arrowheads in (e). No legend for (h).

We added cresyl violet-stained images of the cortex and subiculum from wild type control (Fig. 1e, f).

Legends and explanations were added.

(3) Figs 2 and 4: Expression levels of synaptophysin and phosphorylated tau were evaluated using immunohistochemistry, which is not suitable for quantitative analysis. They should apply another quantitative method such as immunoblotting.

To further add a quantitative method demonstrating the effect of the treatment on disease pathology, we now focused on aggregated tau (Fig. 4e) in the DM-hTAU model and on inflammation (Fig. 3j-m).

(4) Page 7, lines 193-195: DM-hTAU transgenic mouse is a model of FTD but not of AD.

Corrected

(5) Typos;

Page 3, line 75; predisposition

Page 17, line 486; 5 weeks weeks after ...

Corrected

Reviewer #2 (Remarks to the Author):

This study by Rosenzweig.N et al. aim to determine whether PD-1/PD-L1 immune checkpoint blockade could reduce cognitive loss and pathology in a mouse model of tau associated dementia, which further prove the potential application of PD-1/PD-L1 treatment for AD and other neurodegenerative diseases. In general, the experiments are convincing in term of the experiments were well-performed and controlled. However, there are several conceptual and experimental concerns detailed below.

Major points:

1. In figure 1e and 1f, besides neuronal loss showed in the subiculum, images of the cortex are also needed to comprehensively demonstrate that PD-1 blockade treatment could reduce neuronal loss in the brains. Furthermore, it would be more convincing to include WT control mice since neuronal survival in 5XFAD mice were calculated relative to WT mice. For figure 1g, the authors need to provide larger field of images vision.

We added results of imaging and quantification of neuronal survival in the cortex (now Fig. 1h). We also now include an image of a sample from a WT mouse (Fig. 1e).

2. In figure 2j, the authors only showed the changes of il-12p40/ il-10 ratio. It is more accurate to show the relative expression level of il-12p40 and il-10 separately. In addition, the authors should measure the expression of other pro-inflammatory cytokines, i.e. tnf- α and il-6. The reduction of these proinflammatory cytokines would be better evidence for modulation of neuroinflammation response by PD-1/PD-L1 blockade in these mice.

As requested, we now separately show the expression levels of each cytokine (Fig. 2j, k). In addition, in the DM-hTAU model, we further analyzed IL-1 β at the protein levels, and by immunohistochemistry (Fig. 3j-m).

3. As to RAWM experiments in figure 1 and figure 2, it is not clear how the authors determined a mouse with motor deficit and how many mice had motor deficits. Are there any differences between treated and untreated mice? It is necessary to provide additional data to

demonstrate that the motor ability of mice analyzed in RAWM experiments is same among different groups.

Animals with a motor deficit could not be assessed for cognitive performance as the tests that we used require uniform motor activity. We now addressed this issue in Material and Methods and Results. Importantly, analyzing the pathology in all animals, including those with motor defects showed the same pattern (Figures S2, S3).

4. In figure 3, it is not clear what ages of animals were used in the T maze and Y maze behavioral tests? Since the spatial memory ability of both female and male mice were measured using these two behavior test, it is curious about how the female mice behave in Y maze, which was not showed by the authors.

As requested, we now separately show also the results of female and male animals in the behavioral tests (Figure S4).

5. For figure 4e, the authors have previously demonstrated that increased dose of PD-L1 blockade had little effect on enhancing cognitive performance in 5XFAD mice, showed in figure 2b. So it is not significant and might be removed to supplementary data. Additionally, according to similar results, it seems that antibody efficacy is unlikely the dominant factor for treatment in different animal models, which discussed by the authors in line 344-352.

The relationship between antibody dosing and efficacy is now discussed more extensively (Pages 17-18). The dosing is very critical, as a dose of 0.1 mg is ineffective, tested now by using two different anti-PD-L1 antibodies (Fig.3i,n).

6. In figure 5a, effector memory T cells (CD44⁺CD62L^{low}) are abbreviated as “TEM”, but not “TEF”. Are these representative figures from PD-L1 treated mice? It would be more convincing to include WT and IgG isotype control groups. It would be also important to include absolute cell count of lymphocyte populations in Figure 5b and 5d, since cell frequency cannot accurately reflect changes in the cell numbers. In the BM chimeric experiment, it is not clear whether part of CD45⁺CD11b⁻ cells are GFP⁺, which indicate the infiltrating lymphocytes (including T cells or B cells). If so, further phenotypic and functional analysis of this cell subset and monocyte-derived macrophages would probably reflect how systemic PD-1/PD-L1 blockade affect neuroinflammation and pathological process in the brain.

We corrected the abbreviation from TEF to TEM. In the revised manuscript (Fig. 5b), we further analyzed the phenotype of the infiltrating monocyte-derived macrophages (Fig. 5h, i). With respect to the he referee’s question regarding the additional BM-derived GFP⁺ cells; in the flow cytometry analysis we did not include GFP⁺CD45⁺CD11b⁻ cells in our gating strategy (Fig. 5e). In our immunohistochemical analyses, we realized that most of these other BM-derived GFP⁺CD45⁺CD11b⁻ cells are in the meninges and the choroid plexus but not in the parenchyma. We now discuss this issue.

7. Although the results shown in this manuscript and recent work (Nature medicine, 2016) are compelling, a significant conceptual concern from this reviewer is the difference between the findings of this group compared to the findings of Latta-Mahieu, M. et al. (Glia, 2017). Different antibody efficacies discussed by the authors in the text might not the only reason

account for the contradictory results. At least in part, CNS inflammation at preclinical and clinical AD stage could serve different roles in disease progression. To move the field forward, it is important to either perform additional experiments and/or discuss the potential reasons in depth.

Additional experiments were performed using additional antibodies (Fig. 3n). We agree with the referee that to move the field forward the failure described in the paper appearing in *Glia* (cited) requires further discussion, which we now added (Pages 17-18).

Minor points:

1. In the text and figure1: should be “NeuN” but not “Neu-N”.
Corrected.

2. Line 151-152: the description of PD-L1 expression is not correct for its expression in a wide variety of cell types.

Corrected.

3. Line 193 should be rephrased. Transgenic mice with FTD-associated tau mutations are not considered as AD mouse models.

Corrected.

4. Line 258: should be “CD62L” but not “CD62-L”.

Corrected.

5. Line 358-359: Please explain why antibodies were administered once a month? Why not a longer or shorter interval?

An explanation of our choice of dosing interval was added.

6. Line 416: missing closing parenthesis after text “#SIG-39320”.

Corrected

7. Line 867: effector memory T cells should be abbreviated as “TEM”

Corrected

Reviewer #3 (Remarks to the Author):

Based on observations that the recruitment of monocyte-derived macrophages to the brain is dependent on systemic availability of IFN γ -producing CD4⁺ve T cells, the authors speculated that increasing the number of IFN γ -positive cells systemically or at the level of the choroid plexus would promote the recruitment of monocyte-derived macrophages to modulate brain inflammation. They previously demonstrated the benefits of this approach in neurodegenerative disease models by reducing the numbers of FoxP3 regulatory T cells or by blocking the PD-1 immune checkpoint in the 5xFAD mouse model of amyloid pathology. They extend this work here to show that targeting PD-1 or PD-L1 ligands with different treatment regimes shows benefit in 5xFAD mice and in a human tauopathy mouse model. In

the tau mice, administration of an anti-PD-L1 antibody increased monocyte-derived macrophages in the brain, mediated local inflammatory responses and reduced tau phosphorylation and cognitive dysfunction. This is an interesting study describing a new approach to ameliorate neurodegenerative disease by modulating systemic inflammation. The novelty of the work is a little reduced because of previous work in this area by the authors, but this paper does show that the approach has utility in distinct models of human neurodegeneration linked to Alzheimer's disease. The writing and data are high quality, the work is likely to still be of interest to the field, and methods used and statistical analysis performed appear appropriate.

Comments:

1. Nature journals recommend following the ARRIVE reporting guidelines when documenting animal studies (PLoS Bio 8(6), e1000412,2010). This includes reporting the effects of treatment/modifications in wild-type in addition to experimental animals. These controls are missing from the current manuscript which means that it is not possible to ascertain if the effects of treatment are related to disease development in the transgenic mice. The background strain on which DM-htau mice are bred should also be stated.

We now added results testing the effect of treatment in WT mice (Figure S5).
The genetic background of the mice is now indicated

2. Throughout the manuscript, the DM-htau mice are referred to as a mouse model of AD. It would be more accurate to call these a mouse model of tauopathy since the tau mutations expressed by these mice are not linked with AD (no tau mutations are) and as the author correctly states these mutations instead cause FTD.

The reference to the mouse model is now corrected.

3. Fig. 1E. Neuronal survival should be reported for the hippocampus in addition to the subiculum since this is a more meaningful measurement in terms of cognitive decline in the 5xFAD mice.

Quantitation of neurodegeneration in 5XFAD using, cresyl violet, was measured in the cortex and in the subiculum (see Eimer et al., Molecular neurodegeneration, 2013). Results of neuronal survival in the cortex were added (Fig. 1h)

4. Line 144-145 states that pathological analysis was carried out on only animals without a motor impairment. Presumably animals with motor problems were also excluded from the behavioural analysis? This should be more clearly stated. Related to this, why were only 6 WT animals assessed by cresyl violet? (line 774).

The number of animals stained with Cresyl violet is now better explained (figure legend 1).

5. Fig. 2H-J. The conclusion that anti-PD-1/anti-PD-L1 protects synapses would be better demonstrated by immunolabelling with both pre- and post-synaptic markers to illustrate synaptic pairs that are likely to be functional and/or by visualisation and counting of synaptic spines in these groups of animals.

We agree with the referee's comment. We toned-down the conclusion of the results of the experiment, in which we described the effect of the treatment on Synaptophysin (Page 14).

6. The analysis of tau changes in the DM-htau mice following treatment should be expanded to make these findings more compelling. I would prefer to see some biochemical analysis of changes in tau phosphorylation relative to total tau amounts for e.g. by immunoblotting. It would also be useful to examine which species of tau is affected, for example by extraction and quantification of sarkosyl-insoluble tau. It would be useful to explore the pathways that are implicated in these changes by either showing elevation in specific kinases or tau aggregate clearance mechanisms. In addition, the relationship between tau and synapse health should be examined, as described above for analysis of synapses in the 5xFAD mice.

To provide quantitative data showing the effect of the treatment on disease pathology, we now added results showing extraction of the brain proteins and biochemically measured levels of aggregated tau (Fig. 4e) and levels of the inflammatory cytokine IL-1 β (Fig. 3l). In addition, to get further insight to the mechanism, we now explored the phenotype of the infiltrating monocyte-derived macrophages; we found that they exclusively express scavenger receptor that is not expressed by resident microglia (Fig 5. i, k-l), and the anti-inflammatory cytokine (Il-10) that was not found to be expressed by any other cells in the brain (Fig. 5h). Moreover, the infiltrating cells are mainly found in the vicinity of pathological tau (Fig. 5j, l).

7. Discussion: Please include some discussion about how anti-PD-1/anti-PD-L1 treatment is likely to affect astrocyte activation (Fig. 2d). Is it assumed that this results from lowered plaque burden?

Astrocyte were analyzed in Tau pathology (Fig. 3x)

8. Discussion: Please discuss recent papers from the AD field which have genetically modified IL-10 in APP transgenic mice (Chakrabarty et al., 2015 Neuron; Guillot-Sestier et al., 2015 Neuron) and that show effects of modulating this cytokine in terms of microglial responses and plaque burden distinct from the anti-inflammatory role of this cytokine. This is an important body of work to consider in light of the changes shown in this current manuscript.

More results were added regarding inflammation (Fig. 3 and 4), in addition to further thorough discussion with respect to the cited papers (Page 15).

typos

Line 106. Cohort

Corrected

REVIEWERS' COMMENTS:

Reviewer #1 (Remarks to the Author):

The manuscript has been improved. Some typos should be corrected.

Reviewer #2 (Remarks to the Author):

This paper by Rosenzweig.N et al. claims that PD-1/PD-L1 immune checkpoint blockade could reduce cognitive loss and pathology in mouse models of A β and tau associated dementia, which through modulating neuroinflammation by infiltrated monocyte-derived macrophages. Although some details remain uncertain, the authors have addressed most of my concerns. Generally, the results are convincing and interesting. Based on previous study, this work further indicate that immune checkpoint therapy could serve as a promising strategy for treatment of AD and other neurodegenerative disease. Since inflammation (including CNS and peripheral system) may be the early cause of neurodegeneration, depicting the roles of peripheral immune system and CNS immune cells in the process of neurodegeneration is essential to move the field forward.

Reviewer #3 (Remarks to the Author):

The authors have revised the manuscript to address some of the main concerns from the reviewers. The work is well performed and will be of significant interest to the field. Some (minor) issues that remain:

1. I can understand that only one IgG control group is shown in Figures, but I am not convinced that it is valid to combine the data from these two groups for statistical analysis. They are different conditions.
2. The wild-type control line used in each study needs to be clearly stated in the text (results section) since the 5xFAD and DMtau mice are on different genetic backgrounds.
3. Fig. S2 and S3 seem to be unnecessary. They don't add anything to the story.

Point-by-point response to Referee #3

The authors have revised the manuscript to address some of the main concerns from the reviewers. The work is well performed and will be of significant interest to the field. Some (minor) issues that remain:

1. I can understand that only one IgG control group is shown in Figures, but I am not convinced that it is valid to combine the data from these two groups for statistical analysis. They are different conditions.

The two controls were combined in the two figures (Figure 2c, and Figure 3c), because they gave the same results. To demonstrate their equivalence, we added a Supplementary Figure, in which the IgG control groups in Figure 3c are now separately analyzed (Supplementary Fig. 4). Throughout the entire manuscript, we are now referring to the specific isotype control that we used.

2. The wild-type control line used in each study needs to be clearly stated in the text (results section) since the 5xFAD and DMtau mice are on different genetic backgrounds.

This is now described in the Methods, and emphasized throughout the entire manuscript.

3. Fig. S2 and S3 seem to be unnecessary. They don't add anything to the story.

Since the two other referees did not comment on these figures, we decided not to delete them.

Throughout the manuscript, typos were corrected.